# aPKC controls endothelial growth by modulating c-Myc via FoxO1 DNA-binding ability

Meghan Riddell[1], Akiko Nakayama[2], Takao Hikita[1], Fatemeh Mirzapourshafiyi[1], Takuji Kawamura[1], Ayesha Pasha[1], Mengnan Li[1], Mikio Masuzawa[3], Mario Looso[4], Tim Steinbacher[5], Klaus Ebnet [5], Michael Potente[6], Tomonori Hirose[7], Shigeo Ohno [7], Ingrid Fleming[8], Stefan Gattenlöhner[9], Phyu P. Aung [10], Thuy Phung[11], Osamu Yamasaki[12], Teruki Yanagi[13], Hiroshi Umemura[12] & Masanori Nakayama [1]

Strict regulation of proliferation is vital for development, whereas unregulated cell proliferation is a fundamental characteristic of cancer. The polarity protein atypical protein kinase C lambda/iota (aPKCλ) is associated with cell proliferation through unknown mechanisms. In endothelial cells, suppression of aPKCλ impairs proliferation despite hyperactivated mitogenic signaling. Here we show that aPKCλ phosphorylates the DNA binding domain of forkhead box O1 (FoxO1) transcription factor, a gatekeeper of endothelial growth. Although mitogenic signaling excludes FoxO1 from the nucleus, consequently increasing c-Myc abundance and proliferation, aPKCλ controls c-Myc expression via FoxO1/miR-34c signaling without affecting its localization. We find this pathway is strongly activated in the malignant vascular sarcoma, angiosarcoma, and aPKC inhibition reduces c-Myc expression and proliferation of angiosarcoma cells. Moreover, FoxO1 phosphorylation at Ser218 and aPKC expression correlates with poor patient prognosis. Our findings may provide a potential therapeutic strategy for treatment of malignant cancers, like angiosarcoma.

---

[1] Laboratory for Cell Polarity and Organogenesis, Max Planck Institute for Heart and Lung Research, 61231 Bad Nauheim, Germany. [2] Department of Pharmacology, Max Planck Institute for Heart and Lung Research, 61231 Bad Nauheim, Germany. [3] Department of Molecular Diagnostics, School of Allied Health Sciences, Kitasato University, Kanagawa 252-0374, Japan. [4] Bioinformatics Service Group, Max Planck Institute for Heart and Lung Research, 61231 Bad Nauheim, Germany. [5] Institute-Associated Research Group: Cell Adhesion and Cell Polarity, Institute of Medical Biochemistry, ZMBE, University of Münster, 48149 Münster, Germany. [6] Angiogenesis and Metabolism Laboratory, Max Planck Institute for Heart and Lung Research, 61231 Bad Nauheim, Germany. [7] Department of Molecular Biology, Yokohama City University School of Medicine, Yokohama 236-0004, Japan. [8] Institute for Vascular Signaling, Centre for Molecular Medicine, Goethe University, 60596 Frankfurt, Germany. [9] Institute of Pathology, University Hospital, Justus-Liebig-University Giessen, 35392 Giessen, Germany. [10] Department of Pathology University of Texas, M.D. Anderson Cancer Center, Houston, TX 77030, USA. [11] Department of Pathology & Immunology, Baylor College of Medicine, Texas Children's Hospital, Houston, TX 77030, USA. [12] Department of Dermatology, Okayama University, School of Medicine, Okayama 700-8558, Japan. [13] Department of Dermatology, Hokkaido University Graduate School of Medicine, Sapporo 060-8638, Japan. Correspondence and requests for materials should be addressed to M.N. (email: masanori.nakayama@mpi-bn.mpg.de)

Cell proliferation is tightly controlled during development and in tissue homeostasis, while unrestrained cell division is a hallmark of cancer[1,2]. With stimulation by growth factors, such as vascular endothelial growth factors (VEGFs), endothelial cells (ECs), the cells that line the innermost layer of the vasculature, expand rapidly in a tightly coordinated manner to form new vessels[2–4]. Conversely, aberrant EC proliferation is a driver of numerous diseases and occurs in multiple forms of vascular tumors, including angiosarcoma, a malignant vascular neoplasm[5].

Forkhead box O1 (FoxO1), an effector of the phosphatidylinositol-3-OH kinase (PI3K)/Akt pathway, is a key transcriptional regulator of cell proliferation under the control of the receptor tyrosine kinase signaling pathway[6]. Recent work has highlighted that endothelial growth is regulated by FoxO1 downstream of VEGF-A in a context dependent manner[7,8]. VEGF/PI3K/Akt signaling promotes FoxO1 cytoplasmic localization, resulting in its inactivation[8]. Cytoplasmically localized FoxO1 was associated with c-Myc expression and EC proliferation, and loss of FoxO1 resulted in increased EC proliferation[8]. Another work has shown that VEGF-induced EC proliferation is, instead, suppressed with loss of FoxO1. They also found that constitutively active FoxO1 does not inhibit EC proliferation in the liver and the kidney at the adult stage, but leads to lethality due to heart defects[7].

Cell polarity is a fundamental feature of many cells types that is required for proper tissue function. Conversely, loss of polarity causes tissue disorganization and excessive cell growth[1,9]. One of the key regulators of cell polarization, conserved from worms to mammals, is atypical protein kinase C (aPKC)[10]. Disrupted aPKC exhibits not only polarization defects but also altered cell proliferation in Drosophila and Xenopus models[11,12]. In mammals, aPKCλ is often over-expressed and mis-localized in highly malignant tumors, including ovarian, breast, and lung cancer[13–16]. In ECs, loss of aPKCλ leads to hyper-activation of VEGF signaling but, paradoxically, knockout (KO) mice show impaired EC proliferation[17]. However, the molecular mechanism connecting aPKC to cell proliferation remains elusive. Here we provide mechanistic insight into how aPKC regulates endothelial growth. Our study reveals that aPKC controls physiological and pathological vascular growth by regulating the transcriptional activity and abundance of key transcription factors FoxO1 and c-Myc. Moreover, we show that abnormal aPKC/FoxO1/c-Myc signaling contributes to excessive EC proliferation in angiosarcoma.

## Results

**aPKCλ controls c-Myc expression via FoxO1.** Although aPKCλ is a negative regulator of VEGF signaling, loss of aPKCλ in ECs results in decreased proliferation[17]. To begin to understand this conundrum, we examined the expression of FoxO1 and c-Myc in the retinal vasculature at postnatal day 6 (P6) in control and EC specific inducible aPKCλ loss of function (Prkci[iΔEC]) mice (Fig. 1a, b). As previously shown, reduced vascular branching was confirmed in Prkci[iΔEC] mice[17]. FoxO1 expression and localization were indistinguishable between control and Prkci[iΔEC] mice (Fig. 1a). In control animals, a strong c-Myc signal was detected just behind the angiogenic front and in ECs surrounding and within the vein, where proliferation actively occurs[18]. However, c-Myc expression in Prkci[iΔEC] mice was strongly suppressed (Fig. 1b). Additionally, we confirmed that c-Myc expression was also suppressed in human EC after siRNA mediated knockdown of PRKCI (Supplementary Fig. 1a). We have previously reported that a gradient of aPKC activity can be observed in the P6 retinal vasculature, with the highest activity of aPKC observed in the vascular plexus[17]. Consistent with our previous report, there was

no signal corresponding to active aPKC (phospho-aPKC) detected in the tip cells of the angiogenic front, but a jump in the activity of aPKC could be seen in the EC just behind the leading edge of the vascular front, where c-Myc was abundantly expressed (Supplementary Fig. 1b). The strongest signal for activated aPKC was observed in the more mature vessels of the vascular plexus (Supplementary Fig. 1b). Nuclear localized FoxO1 was also most strongly observed in the vascular plexus compared to the angiogenic front (Supplementary Fig. 1c). To confirm the effect of aPKC deletion on c-Myc expression just behind the angiogenic front, we carried out mosaic deletion experiments using an EYFP Cre reporter mouse line. After mosaic deletion of aPKC due to a single low dose injection of tamoxifen at P1, c-Myc signal was significantly reduced in aPKC deficient cells expressing the EYFP Cre reporter compared to that of non-recombined control EYFP negative cells within the same retina (Supplementary Fig. 1d, e).

To examine whether the observed alteration in c-Myc expression upon aPKCλ deletion is mediated via FoxO1, we generated endothelial specific inducible Prkci/Foxo1 double KO mice. These double KO mice displayed a striking phenotype characterized by increased vascular coverage at the distal ends and associated angiogenic front of only the venous but not arterial regions (Fig. 1c, d; Supplementary Fig. 2a, b). Consistent with previous reports, Prkci mutants displayed reduced, and Foxo1 mutants displayed increased venous vascular coverage (Fig. 1d; Supplementary Fig. 2a)[8,17]. Double KO venous vascular coverage was not significantly different from Foxo1 mutants but significantly increased compared to Prkci mutants (Fig. 1d; Supplementary Fig. 2a). Interestingly, relative radial expansion of the retinal vasculature was modestly decreased in Prkci[iΔEC] mutants, but strongly reduced in FoxO1 [iΔEC] and double KO mice without showing an additive effect (Fig. 1e; Supplementary Fig. 2a). Additionally, distinct stumpy hyperdense vascular regions were observed in ~75% of veins and their associated angiogenic front of the double knockout retinas (Fig. 1c; Supplementary Fig. 2a, c), and were also observed in the Foxo1 mutant retina (Supplementary Fig. 2a, c)[8]. This phenotype displayed an equal degree of penetrance in both double KO and Foxo1[iΔEC] mutants (Supplementary Fig. 2a, c).

We confirmed that c-Myc expression was upregulated in the retinal angiogenic front of the EC specific FoxO1 mutant mice (Fig. 1g), and, importantly, the impaired c-Myc expression observed in the Prkci[iΔEC] mutant retina EC was rescued with double KO of Prkci and Foxo1 (Fig. 1f, g). These results suggest that aPKC controls c-Myc expression via FoxO1 through a mechanism that does not involve FoxO1 sequestration to the cytoplasm, as observed during VEGF signaling.

**aPKC phosphorylates FoxO1 at Ser218.** To understand how aPKCλ affects c-Myc expression via FoxO1, we investigated whether aPKCλ can phosphorylate FoxO1. Purified full-length FoxO1 was effectively phosphorylated when incubated with recombinant aPKCλ (Fig. 2a). To identify aPKCλ-dependent phosphorylation sites in FoxO1, phosphorylated FoxO1 was analyzed using a phosphoproteomic approach. Mass spectrometry analysis identified serine 218 (Ser218) as the predominant phosphorylation site in FoxO1, conserved from worms through all mammalian FoxOs and located within its DNA-binding domain (DBD) (Supplementary Fig. 3a, b). Mutating Ser218 to Ala almost completely abolished FoxO1 phosphorylation by aPKCλ in vitro, confirming that Ser218 is the major phosphorylation site (Fig. 2a). To examine the phosphorylation of FoxO1 at Ser218 in vivo, we generated two antibodies that specifically recognize Ser218-phosphorylated (pSer218) FoxO1. The specificity of the antibody was assessed by western blot analysis. GST-FoxO1 DBD phosphorylated by purified aPKCλ in vitro was specifically detected by anti-pSer218 antibodies in a dose

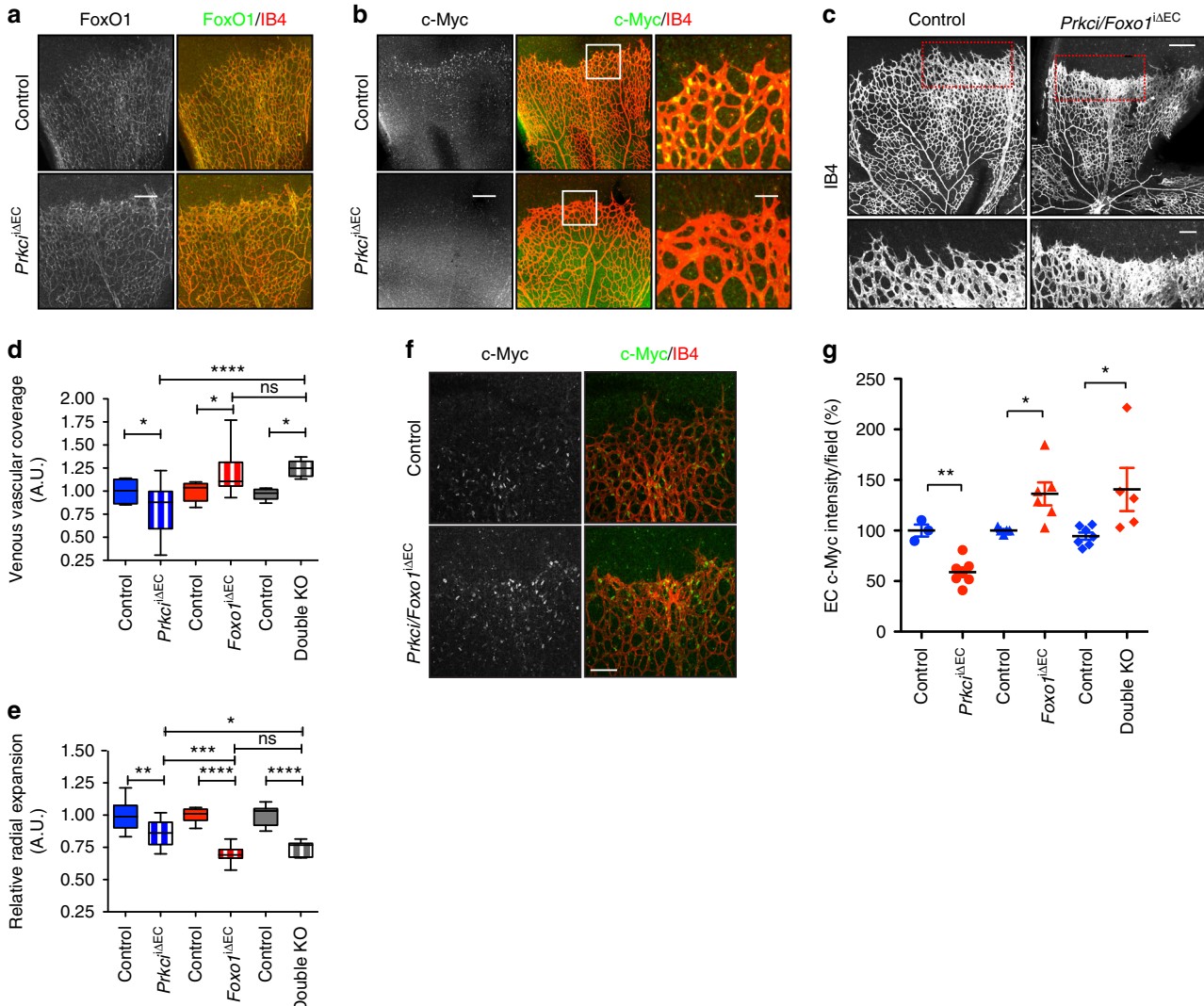

**Fig. 1** aPKCλ controls c-Myc abundance via FoxO1. **a** Staining of FoxO1 and isolectin-B4 (IB4) in *Prkci*$^{i\Delta EC}$ and control mouse retina at postnatal day (P)6. Staining is representative of 3 animals of each genotype; Scale bar represents 200 μm. **b** Staining of c-Myc and IB4 in *Prkci*$^{i\Delta EC}$ and control mouse retina at P6. Higher magnification images of indicated areas presented on the right; Staining is representative of 3 control and 7 KO retinas; Scale bar in the left panel represents 200 μm. Scale bar in the higher magnification image represents 50 μm. **c** IB4 staining of *Prkci/Foxo1*$^{i\Delta EC}$ and control mice at P6. Bottom panel is higher magnification image of indicated area from the upper panel of each genotype. Images are representative of >10 animals of each genotype; Scale bar in the upper panel and the lower panel represents 250 μm and 100 μm, respectively. **d** Quantification of relative ratio of vascular coverage in the venous region; line represents median, whiskers representing range; one-way ANOVA Bonferroni post-hoc analysis; *$p < 0.05$; ****$p < 0.0001$; n.s $p > 0.05$; ($n \geq 10$). **e** Quantification of relative radial expansion of the retinal vasculature at P6; line represents median, whiskers representing range; one-way ANOVA Bonferroni post-hoc analysis; n.s. $p > 0.05$; *$p < 0.05$; **$p < 0.01$; ***$p < 0.001$; ****$p < 0.0001$; ($n \geq 10$). **f** c-Myc and IB4 staining in *Prkci/Foxo1*$^{i\Delta EC}$ and control P6 retina. Staining is representative of >8 animals of each genotype; Scale bar represents 100 μm. **g** Quantification of c-Myc signal intensity of P6 vascular front; Data represent mean ± S.E.M. two-tailed unpaired *t*-test *$p < 0.05$; **$p < 0.01$ ($n \geq 3$)

dependent manner (Supplementary Fig. 3c). Ectopic expression of aPKCλ in cultured ECs resulted in a marked increase in the reactivity of the pSer218-FoxO1 antibodies (Fig. 2b), whereas treatment of the cells with a myristoylated aPKC pseudosubstrate inhibitor (aPKC kinase inhibitor) diminished pSer218 reactivity (Fig. 2b). We next used these antibodies to analyze Ser218 phosphorylation of FoxO1 during angiogenesis. Consistent with our analyses of aPKC activity in the retinal vasculature, pSer218-FoxO1 immunofluorescent signals detected with both antibodies were below detectable levels in the tip cells of the angiogenic front, but a signal could be detected in the EC just behind the leading edge of the angiogenic front, and the strongest signal was observed in the more established and remodeled vessels of the central retina (Fig. 2c; Supplementary Fig. 3d). On the other hand, immunoreactive signal against total FoxO1 distributed

evenly throughout the whole retinal vasculature (Fig. 1a; Supplementary Fig. 1c). Moreover, signal corresponding to pSer218 in FoxO1 was strongly decreased in *Prkci*$^{i\Delta EC}$ mice and *Foxo1*$^{i\Delta EC}$ mice (Fig. 2d, e). The immunofluorescent signal for pSer218-FoxO1 in the vascular plexus of P6 retina was observed in both cytoplasmic and nuclear compartments (Fig. 2f). Taken together, we conclude that aPKCλ phosphorylates FoxO1 at Ser218 in ECs in vitro and in vivo.

**aPKCλ controls the DNA-binding ability of FoxO1 but not its localization.** The PI3K/Akt signaling pathway controls the nuclear localization of FoxO1 via phosphorylation at Thr24/Ser256/Ser319, which is believed to be a major FoxO1 regulatory mechanism[6]. To determine whether the phosphorylation of

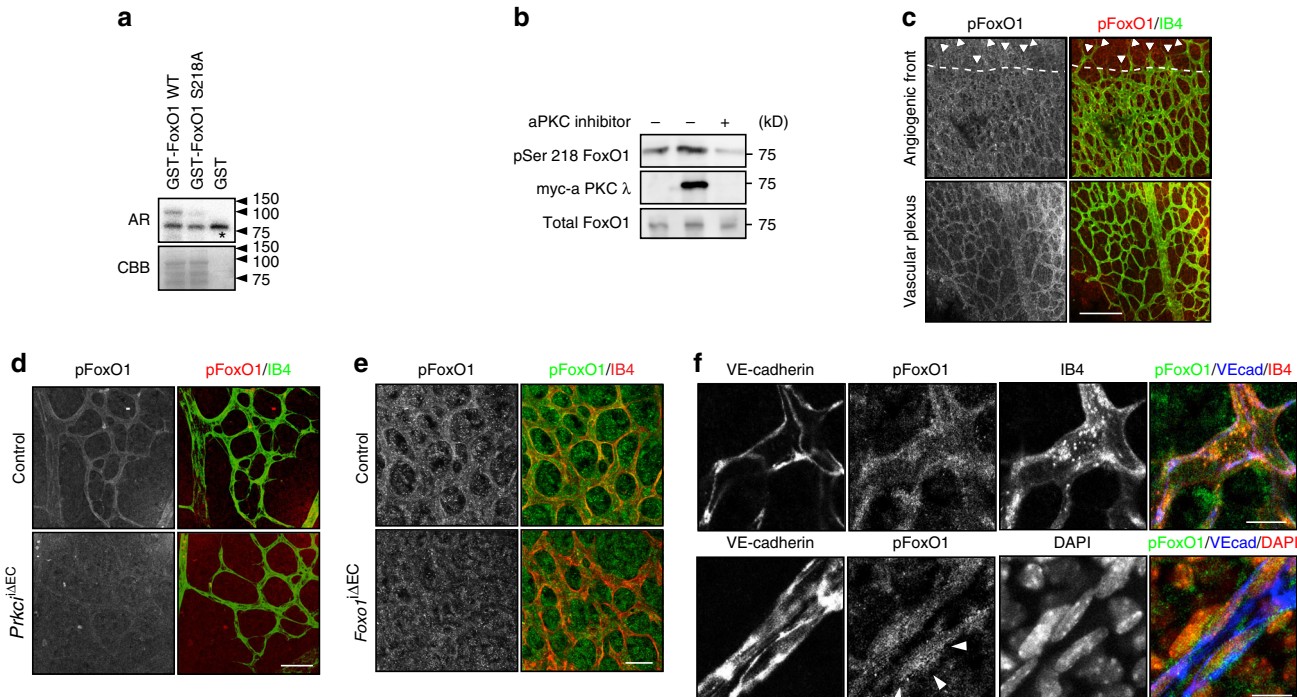

**Fig. 2** aPKCλ phosphorylates FoxO1 at Serine 218. **a** Phosphorylation of FoxO1 by aPKCλ in vitro. Radioactive proteins were detected by autoradiography (AR). Autophosphorylation of aPKCλ is indicated with asterisk. Total protein is shown with Coomassie Brilliant Blue (CBB). **b** Western blot detection of pSer218 with an anti-pSer218-FoxO1 antibody in cultured ECs overexpressing aPKCλ with and without aPKC inhibitor treatment. **c** Staining of pSer218 using an anti-pSer218-FoxO1 antibody and isolectin-B4 in P6 control mouse retina. Arrowheads indicate the position of tip cells and dashed line indicates approximate threshold where pSer218-FoxO1 is detectable; Staining is representative of 6 animals examined; Scale bar represents 100 μm. **d** Staining of pSer218 using an anti-pSer218- FoxO1 antibody and IB4 in Prkci[iΔEC] and control P6 retina vascular plexus. Staining is representative of 3 animals of each genotype; Scale bar represents 50 μm. **e** Staining with anti-pSer218-FoxO1 antibody and IB4 in Foxo1[iΔEC] and control P6 retina vasculature. Staining is representative of 3 animals of each genotype; scale bar represents 25 μm. **f** Phosphorylation of FoxO1 at Ser218 in control P6 retina vascular plexus detected with anti-pSer218-FoxO1, anti-VE-cadherin, DAPI (lower panels), and IB4 (upper panels). Staining is representative of 6 animals examined; Scale bar represents 10 μm

FoxO1 by aPKCλ controls its nuclear localization, we separated the nuclear and cytosolic compartments from ECs. Consistent with the in vivo observation in mouse retina, KD of *PRKCI* did not affect the sub-cellular localization of FoxO1 in cultured ECs (Supplementary Fig. 4a). Furthermore, overexpression of aPKCλ did not have any effect on the nuclear localization of FoxO1 (Supplementary Fig. 4b). Active aPKCλ localized only within the cytosol (Supplementary Fig. 4b).

To gain insight into the consequences of the phosphorylation of FoxO1 by aPKCλ, we looked at the location of Ser218 using the published structure of DNA bound FoxO1[19]. Ser218 is located within the FoxO1 DBD (Fig. 3a), and is predicted to have a critical role in regulating the interaction of FoxO1 with the phosphate backbone of DNA. To confirm this, we substituted Ser218 to aspartic acid to mimic phosphorylation (S218D) and assayed the ability of the mutant to bind DNA using an electrophoretic mobility shift assay. Purified FoxO1 DBD bound to the canonical FoxO DNA-binding element (DBE) as previously reported[19]. However, the FoxO1 S218D DNA-binding domain was unable to bind to this sequence (Fig. 3b). To confirm that the abolition of DNA-binding induced by aPKC dependent phosphorylation has an effect on FoxO1 function in cultured cells, we carried out Luciferase assays in HEK293 cells using the FoxO-responsive 6xDBE luciferase construct. High luciferase activity could be induced after overnight serum starvation (SS) as a result of FoxO nuclear localization. Treatment of the cells with aPKC kinase inhibitor under these conditions significantly enhanced luciferase activity, suggesting that endogenous FoxOs transactivates the 6xDBE reporter more effectively when aPKC is inhibited

(Fig. 3c). Mutation of all the Akt phosphorylation sites into alanine promotes nuclear localization of FoxO1 thereby promoting constitutive activation (FoxO1-CA)[6]. While overexpression of FoxO1-CA induced high luciferase activity of the 6xDBE reporter, FoxO1-CA-S218D (FoxO1-CA/D) did not (Fig. 3d). Given the predominant localization of aPKCλ within the cytosol (Supplementary Fig. 4b), we used wild-type FoxO1 with another luciferase construct containing a 479 bp region of the carboxy-terminal domain RNA polymerase II polypeptide A small phosphatase 2 (CTDSP2) promoter which contains a known FoxO binding site[20]. Overexpression of wild-type FoxO1 was able to enhance luciferase activity without SS, however, co-expression of aPKCλ with FoxO1 suppressed it (Fig. 3e). A reporter construct carrying a point mutation within the FoxO binding sequence of the CTDSP2 promoter region had significantly reduced luciferase activity in all conditions (Fig. 3e). These results suggest that aPKCλ inhibits FoxO1 activity without dynamically affecting its cellular localization.

We further investigated the significance and role of FoxO1 phosphorylation by aPKCλ using chromatin immunoprecipitation. FoxO1 has been shown to interact with target gene promoters in a DNA-binding dependent or independent manner[21]. As such, we used primers to amplify precipitated chromatin from p27, a DNA-binding dependent FoxO1 direct target gene[22], and cyclin D1, an indirect target gene[21]. FoxO1-CA but not FoxO1-CA/D formed a complex with the p27 promoter (Fig. 3f). In contrast, both FoxO1-CA and FoxO1-CA/D were bound to the promoter of cyclin D1 (Fig. 3g). Thus, these results indicate that phosphorylation of FoxO1 by aPKCλ specifically

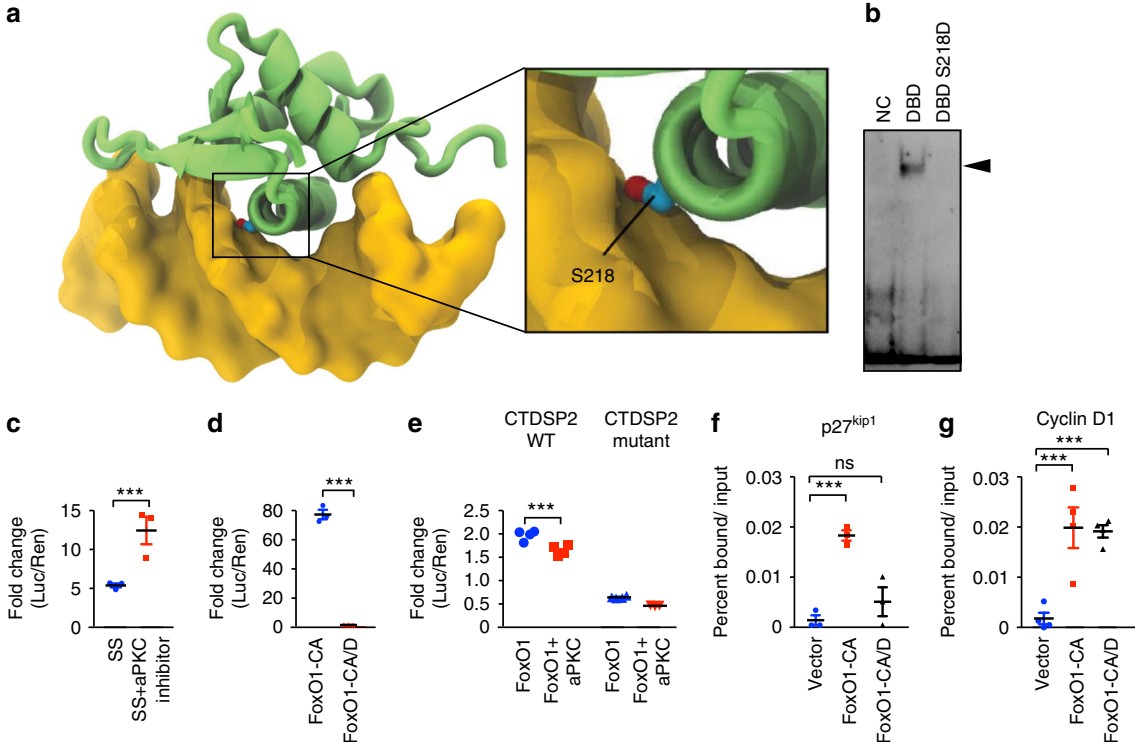

**Fig. 3** aPKCλ regulates FoxO1 DNA binding via Serine 218 phosphorylation. **a** Structural depiction of DNA bound FoxO1 DNA-binding domain. FoxO1 (green) and DNA (yellow). Position of Ser218 indicated. **b** Electrophoretic mobility shift assay with FoxO DBE consensus binding sequence-containing DNA using GST-FoxO1 DBD and GST-FoxO1 DBD S218D. Arrowhead indicates shifted DNA. NC, negative control. **c** The effect of aPKC inhibition on endogenous FoxO1 activity examined with the 6xDBE-luciferase vector. Data represents mean ± S.E.M. two-tailed unpaired $t$-test ***$p < 0.001$ ($n = 3$). **d** FoxO1 activity examined with the 6xDBE-luciferase vector and FoxO1-CA or FoxO1-CA/D mutants. Data represents mean ± S.E.M. two-tailed unpaired $t$-test ***$p < 0.001$ ($n = 3$). **e** The effect of aPKCλ overexpression on FoxO1 activity was examined with CTDSP2-luciferase vectors. The CTDSP2-luciferase construct either contains the WT CTDSP2 promoter sequence (see materials and methods) or the mutant version where a point mutation was introduced in the FoxO binding sequence. Data represents mean ± S.E.M. two-tailed unpaired $t$-test ***$p < 0.001$ ($n = 4$). **f** Chromatin-immunoprecipitation (ChIP) of p27[kip1] DNA-binding-dependent regulatory region with FoxO1-CA or FoxO1-CA/D in HUVEC. Data represents mean ± S.E.M. one-way ANOVA with Bonferroni post-hoc analyses; ***$p < 0.001$; ns $p > 0.05$. ($n = 3$) **g** ChIP of a known Cyclin-D1 DNA-binding-independent regulatory region with FoxO1-CA or FoxO1-CA/D in HUVEC. Data represents mean ± S.E.M. one-way ANOVA with Bonferroni post-hoc analyses ***$p < 0.001$ ($n = 4$)

regulates its DNA-binding ability without affecting FoxO1 dynamic localization and selectively determines the direct interaction of FoxO1 with its target gene promoters.

**aPKCλ phosphorylation of FoxO1 controls c-Myc via miR-34c expression.** We next addressed the molecular mechanism underlying reduced c-Myc expression under aPKCλ loss-of-function conditions. Previous work has established multiple regulatory pathways for the control of c-Myc expression by FoxO transcription factors including the ubiquitin ligase Fbxw7, the transcriptional repressor Mxi1, and two micro-RNAs (miRNAs), miR-34c and miR-145[8,23–27]. The transcription of Fbxw7, Mxi1, and miR-145 was not altered in *PRKCI* KD ECs (Supplementary Fig. 5a, b). In contrast miR-34c expression was significantly upregulated in *PRKCI* KD ECs and in the *Prkci*[iΔEC] mutant retinal vasculature (Fig. 4a, b). The expression of miR-34c was also found to be upregulated after overnight treatment of EC with aPKC kinase inhibitor, indicating that aPKC regulation of miR-34c is kinase activity dependent (Fig. 4c). The promoter region of *MIR34C* contains a FoxO DBE consensus sequence and FoxO binding to this position has previously been shown to induce miR-34c expression[25]. Using chromatin-immunoprecipitation, we confirmed that FoxO1-CA but not FoxO1-CA/D could significantly bind this region of the miR-34c promoter in cultured ECs (Fig. 4d). We then examined whether aPKCλ modulation of c-Myc abundance is controlled by miR-34c. The injection of an anti-miR-34c inhibitor into newborn pups restored the

compromised c-Myc expression and vascular defects observed in P6 *Prkci*[iΔEC] mutant retinal vasculature upon tamoxifen induced gene inactivation (Fig. 4e–g). Consistently, double KO of *Prkci* and *Foxo1* normalized the upregulated miR-34c expression observed in the *Prkci* mutant retina (Supplementary Fig. 5c). Constitutively nuclear FoxO1 has been shown to suppress c-Myc expression as well as genes involved in c-Myc signaling, thereby controlling cell cycle and metabolism in ECs[8]. To gain further insight into the role of aPKCλ in c-Myc signaling, we examined expression of these genes. Gene expression of Cdk4, cyclin-B2, enolase-1, FASN, and LDHB was suppressed in aPKC KD ECs, while expression of genes encoding cyclin-D1, LDHA, cyclin-D2, and PKM2 was not affected (Fig. 4h). Of note, the transcriptional regulation of cyclin-D1, and cyclin-D2 genes by FoxO1, are independent of its DNA-binding activity[21]. These results suggest that loss of aPKCλ selectively affects transcription of target genes that are FoxO1 DNA-binding dependent, including miR-34c and c-Myc-signaling genes.

**aPKCλ/FoxO1 signaling axis in active in angiosarcoma.** aPKCλ, well known as a cell polarity factor, is over-expressed in multiple types of malignant tumors[13,28,29]. Surprisingly, apical basal polarization visualized by immunoreactive signals against anti-podocalyxin and collagen IV antibodies did not exhibit clear differences between control and *Prkci*[iΔEC] mice (Supplementary Fig. 6a). While VE-cadherin and junction adhesion molecule A (JAMA) expression and localization did not show any obvious

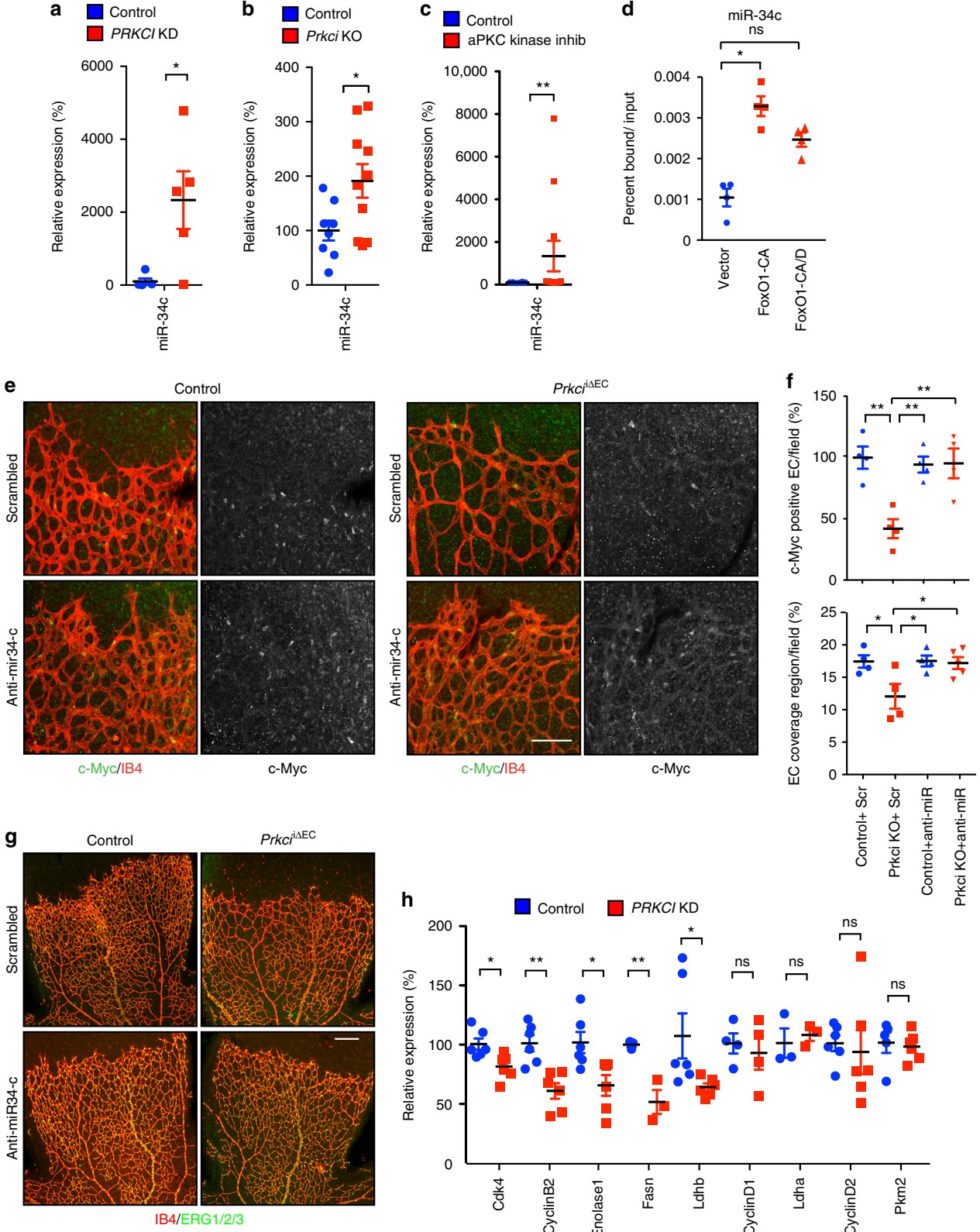

defects in *Prkci*[iΔEC] mutant mice, claudin 5 expression and transcription were compromised in EC specific aPKCλ loss of function mice (Supplementary Fig. 6b-d). Claudin 5 is a transcriptional target of FoxO1 and a marker of angiosarcoma cells within transformed lesions[30,31]. Angiosarcoma is a highly malignant tumor characterized by rapidly proliferating,

extensively infiltrating anaplastic cells derived from ECs. C-Myc has been reported to be over-expressed in angiosarcoma[5], and simultaneous conditional inactivation of all three isoforms of FoxO in mice lead to the development of lethal angiosarcoma[32]. Thus, we hypothesized that the aPKCλ/FoxO1 signaling axis might be strongly activated in angiosarcoma. We performed

**Fig. 4** aPKCλ regulates c-Myc abundance and signaling via miR-34c. **a** Expression of miR-34c in control or *PRKCI* KD siRNA treated HUVEC. Data represent mean ± S.E.M. two-tailed unpaired *t*-test *$p < 0.05$ ($n = 5$). **b** Expression of miR-34c in *Prkci*$^{\Delta EC}$ and control P6 retina. Data represent mean ± S.E.M. two-tailed unpaired *t*-test *$p < 0.05$ ($n \geq 8$). **c** Expression of miR34c after overnight serum starvation with and without aPKC kinase inhibitor treatment in HUVEC; Data represents mean +/- S.E.M., Mann Whitney test; **$p < 0.01$ ($n \geq 8$). **d** ChIP of FoxO1-CA and FoxO1-CA/D from the FoxO1 binding site of the miR-34c promoter region in HUVEC. Data represent mean ± S.E.M. one-way ANOVA with Bonferroni post-hoc analyses; ns $p > 0.05$; *$p < 0.05$ ($n = 4$). **e** Staining of c-Myc (green) and IB4 (red) with c-Myc channel alone in right panels in P6 *Prkci*$^{\Delta EC}$ and control mouse retina after treatment with anti-miR-34c or scrambled control. Representative staining of > 3 animals of each genotype and treatment; Scale bar represents 100 μm. **f** Percentage of c-Myc positive EC cells per field (upper) and endothelial cell coverage region in P6 retina of *Prkci*$^{\Delta EC}$ after treatment with anti-miR-34c or scrambled control. Data represent mean ± S.E.M. two-way ANOVA with Bonferroni post-hoc analyses; **$p < 0.01$; *$p < 0.05$; ns $p > 0.05$ ($n = 3$–5). **g** IB4 (red) and ERG1/2/3 (green) staining in the P6 retina of *Prkci*$^{\Delta EC}$ after treatment with anti-miR-34c or scrambled control. Representative staining of > 3 animals of each genotype and treatment; Scale bar represents 200 μm. **h** Relative expression of FoxO1-regulated genes involved in c-Myc signaling measured by RT-PCR in HUVEC. Data represent mean ± S.E.M. two-tailed unpaired *t*-test ns $p > 0.05$; *$p < 0.05$; **$p < 0.01$ ($n \geq 3$)

immunostaining on EC-derived tumor tissues from cases of primary non-radiation-induced angiosarcoma and, as a non-malignant control, the lobular capillary hemangioma, pyogenic granuloma. Co-immunostaining was carried out with an anti-VE-cadherin antibody, in order to identify the angiosarcoma or pyogenic granuloma cells in transformed lesions[33], along with anti-c-Myc, FoxO1, aPKCλ, or pSer218-FoxO1 antibodies. Consistent with previous studies[5,34], 16 of 18 angiosarcoma samples showed modestly or highly elevated c-Myc expression compared to untransformed cells in the surrounding tissue and in pyogenic granuloma (Fig. 5a, Supplementary Fig. 7a; Supplementary Table 1). It has recently been shown that a cytoplasmic cleavage product of Myc, known as Myc-nick, can be detected by specific c-Myc antibodies, but not the commonly used mouse monoclonal 9E10 c-Myc antibody[35]. Due to the strong cytoplasmic signal observed by immunostaining in the angiosarcoma samples, we performed a western blot on tissue sections from an angiosarcoma patient and confirmed that two bands could be detected by western blotting analysis, with the lower band matching the molecular weight reported for Myc-nick (Supplementary Fig. 7b). Additionally, high-level c-Myc amplification has been observed in radiation-induced angiosarcoma and has also been reported to be a rare event in highly overexpressing c-Myc cases of primary angiosarcoma[36,37]. Of the highly c-Myc overexpressing patients in our study (++ or +++ scoring) a single patient (#11) was found to have c-Myc amplification by FISH. Despite an established mutual antagonism between FoxO1 and c-Myc[8], 14 of 18 angiosarcoma samples showed strong FoxO1 expression, whereas FoxO1 expression in pyogenic granuloma samples was low (Fig. 5b; Supplementary Fig. 7; Supplementary Table 1). Moreover, 14 of 18 angiosarcoma samples showed clear or partial nuclear localization of FoxO1 with strong protein expression (Fig. 5b; Supplementary Fig. 7; Supplementary Table 1). aPKCλ was highly expressed and mis-localized from cell-to-cell contact sites in 24 of 39 angiosarcoma samples, but not pyogenic granuloma samples (Fig. 5c; Supplementary Fig. 7; Supplementary Table 1), a pattern that is often observed in malignant cancers[28]. Importantly, a strong signal corresponding to pSer218-FoxO1 was confirmed in 28 of 39 angiosarcoma samples, but not pyogenic granuloma (Fig. 5d; Supplementary Fig. 7; Supplementary Table 1). Ki67 staining was carried out on 16 patient samples and a Ki67 index was calculated. When classified into Ki67 high (44% (7/16)) and low (56% (9/16)) groups, an association could be found between Ki67 expression, c-Myc ($p = 0.0014$; Chi square test for trend), and aPKC ($p = 0.0271$; Chi square test for trend), but not FoxO1 ($p = 0.0536$; Chi square test for trend) expression level. The angiosarcoma patients were also classified into strong- or weak-aPKC expressing groups and pSer218 positive- or negative-groups, and we compared their survival time using a Kaplan–Meier analyses. The survival time of the low-aPKC expressing group and the pSer218-FoxO1 negative group were significantly longer than that of the high-aPKC and pSer218-FoxO1-positive groups respectively (Fig. 6; Supplementary

Table 1). These results indicate the correlation of FoxO1 phosphorylation at Ser218 by aPKC with angiosarcoma malignancy.

To further link the expression level of aPKCλ and c-Myc with pathologic EC proliferative capacity, we employed two different patient-derived angiosarcoma cell lines, ISO-HAS-B and AS-M[38,39]. Previous work has found that ISO-HAS-B, but not AS-M, cells are tumorgenic when injected into immune-compromised mice[38,39]. Proliferation of these cell lines was measured using BrdU incorporation, and ISO-HAS-B proliferation was elevated compared to primary cultured ECs, whereas that of AS-M was reduced (Supplementary Fig. 8a). Moreover, expression levels of aPKCλ, c-Myc, and pSer218-FoxO1 in ISO-HAS-B were higher than that of primary cultured ECs, while expression of all in AS-M was low (Supplementary Fig. 8b). Additionally, aPKC was mis-localized from EC-EC junction sites in ISO-HAS-B cells (Supplementary Fig. 8c). The anti-rheumatoid agents aurothioglucose (ATG) and aurothiomalate (ATM) block the binding of PAR-6 and aPKCλ thereby impairing aPKCλ activation[40,41]. ATG and ATM inhibit non-small lung cancer growth and are in an ongoing clinical trial[14,40–43]. Pharmacological inhibition of aPKC activity in ISO-HAS-B using ATG, ATM, and aPKC kinase inhibitor suppressed FoxO1 phosphorylation and c-Myc expression in a dose dependent manner, and reduced proliferation (Fig. 7a, b; Supplementary Fig. 8d-f). aPKC KD or treatment with aPKC inhibitor in ISO-HAS-B cells induced miR-34c expression, and the treatment of ISO-HAS-B cells with anti-miR-34c inhibitor induced c-Myc expression and proliferation (Fig. 7c, d; Supplementary Fig. 8g; 9a). In addition, FoxO1-CA mutant expression in ISO-HAS-B cells suppressed c-Myc abundance examined by immunofluorescence, as expected, whereas c-Myc levels in FoxO1-CA/D expressing cells was not significantly different from non-transfected neighboring cells, and significantly higher than that of FoxO1-CA expressing cells (Fig. 7e). Both constructs reduced proliferation in ISO-HAS-B cells compared to non-transfected neighboring cells, though FoxO1-CA/D displayed a significantly increased proliferation compared to the FoxO1-CA mutant (Fig. 7f). Importantly, when ISO-HAS-B were treated with aPKC inhibitor after FoxO1 KD, miR-34c expression was normalized, confirming the requirement of FoxO1 expression for aPKC dependent miR-34c expression (Supplementary Fig. 9a). Unexpectedly, FoxO1 KD alone in ISO-HAS-B cells lead to reduced c-Myc expression and proliferation, and increased miR-34c expression (Supplementary Fig. 9a-c). No additive effects were observed with aPKC inhibitor treatment under these conditions. Thus, it appears that the ability of FoxO1 to bind to DNA is important for regulating miR-34c and c-Myc expression in ISO-HAS-B, but its DNA-binding-independent functions are sufficient to suppress proliferation. Altogether the results in the angiosarcoma patient samples and cell line indicate that high aPKC expression and mislocalization, and the resulting FoxO1 phosphorylation are involved in malignant EC proliferation by controlling c-Myc expression.

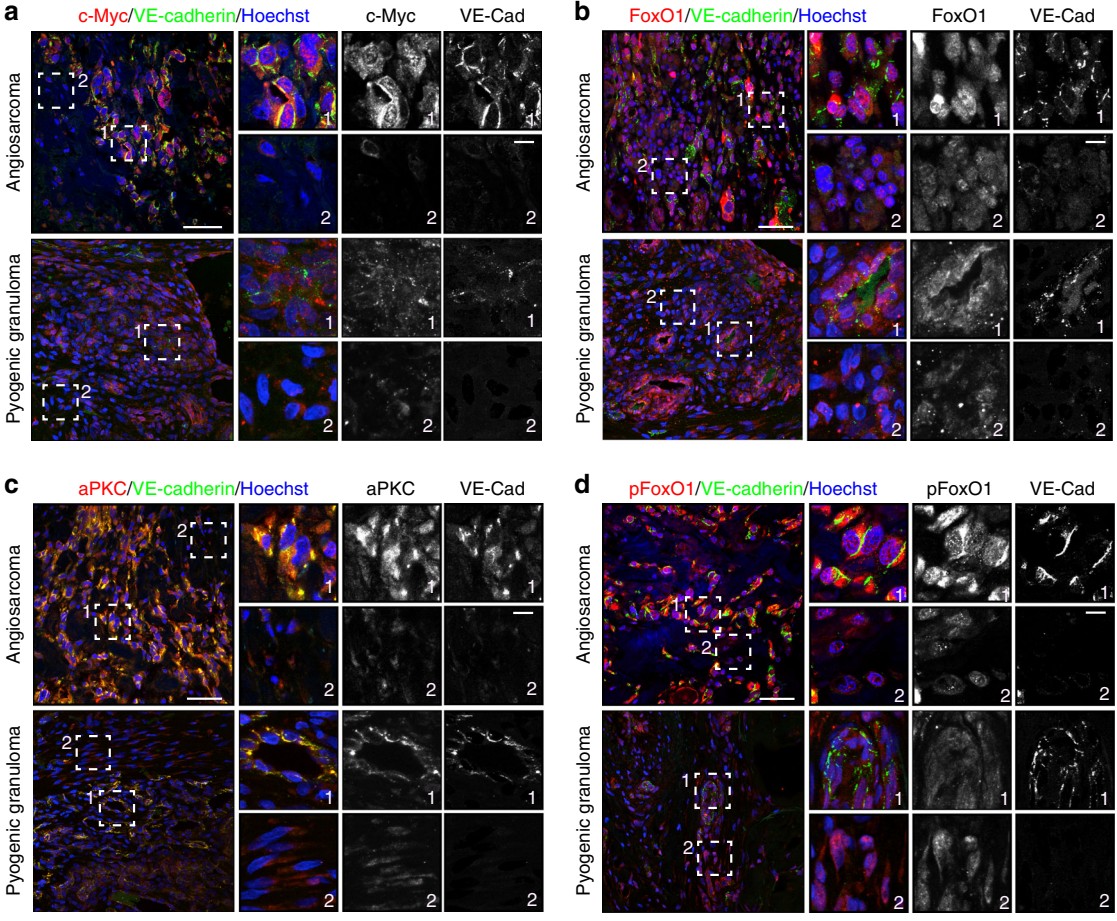

**Fig. 5** aPKCλ is involved in vascular tumor malignancy. **a** Staining of c-Myc, **b** FoxO1, **c** aPKC, or **d** pSer218-FoxO1 (all red channel) and VE-cadherin (green) with Hoechst 33342 (blue) in malignant angiosarcoma (upper panels) and benign pyogenic granuloma (lower panels) tissue samples. Higher magnification fields are indicated with dashed boxes. Box 1 indicates VE-Cadherin positive regions and box 2 indicates VE-cadherin negative regions. Scale bar in the left panels represents 50 μm. Scale bar in the higher magnification field represents 10 μm

## Discussion

Here we found that aPKCλ directly controls FoxO1 activity via phosphorylation in a manner independent of its localization, resulting in modulation of c-Myc expression (Supplementary Fig. 10). We establish that this pathway regulates c-Myc expression during developmental angiogenesis and in malignant endothelial neoplasia. We observed that FoxO1 and c-Myc were highly expressed in the majority of angiosarcoma patient samples, despite known mutual antagonism between these two transcription factors (Supplementary Table 1; Supplementary Fig. 7). Furthermore, aPKCλ expression was high and mis-localized from cell-to-cell junction sites in most of the cases observed. Importantly, FoxO1 Ser218 was highly phosphorylated in many of the patients. Single or double deletion of the three FoxO isoforms, FoxO1, FoxO3, and FoxO4, in mice has been found to result in tissue specific development of hemangiomas, whereas, only simultaneous deletion of all three isoforms resulted in lethal angiosarcoma[32]. Consistent with these observations, the predominant aPKCλ dependent FoxO1 phosphorylation site is conserved in all of the mammalian FoxOs.

Given this critical role of PKCλ activity in FoxO1 regulation, it will be important to determine the relevant regulators that act further upstream. Of the numerous direct and indirect molecular interactions with PKCλ known, the association of the PAR-3/PAR-6/aPKC polarity complex to junctional proteins and, in particular, cadherin family cell adhesion molecules are noteworthy[10]. Junctions in more established vessels have presumably

higher stability and might sustain or recruit more complexes containing active aPKC. Indeed, it has been shown that PKCλ activation is impaired in VE-cadherin loss of function in vitro and in vivo [17,44], suggesting that junction formation and maturation is a trigger of PKCλ activity. However, it has been also reported that the VE-cadherin-associated polarity complex lacks aPKC in cultured ECs[45]. Future work will have to address the connections between aPKC, junctional proteins and other potential regulators.

We demonstrated that angiosarcoma proliferation and c-Myc expression were reduced by treatment with aPKC kinase inhibitors ATG, ATM, and aPKC pseudosubstrate inhibitor. Previous work has found that the aPKC pseudosubstrate inhibitor used in this study can be somewhat non-specific in certain cellular contexts, and is not capable of inhibiting a splice variant of aPKC (PKM)[46]. Despite this, in our experiments, it appears to be working through aPKC for it could reproduce results obtained both with genetic inactivation, siRNA mediated knockdown, and other known aPKC inhibitors.

Our results in the angiosarcoma cell line revealed an unexpected increase in miR-34c expression and decrease in c-Myc expression and proliferation when FoxO1 was knocked down. This may indicate the potential importance of FoxO in angiosarcoma homeostasis. FoxO proteins are known to interact with p53, and this interaction regulates the expression of specific target genes and the activity of both transcription factors[47–49]. P53 is known to regulate miR-34c expression in transformed cells[50] and is frequently mutated and expressed in angiosarcoma[51–53].

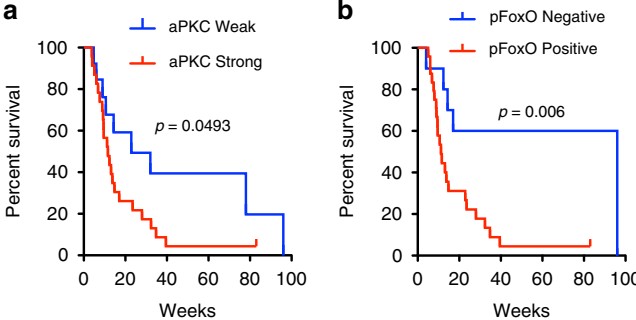

**Fig. 6** aPKC expression level and FoxO1 phosphorylation status in patient survival. **a** Kaplan–Meier survival curve of aPKC and **b** pSer218-FoxO1 expression in angiosarcoma patient samples. Mantel–Cox log rank test; aPKC $\chi^2 = 3.866$, df $= 1$ $p = 0.0493$; pSer218-FoxO1 $\chi^2 = 7.541$, df $= 1$ $p = 0.0060$

Expression of p53 has also been demonstrated in ISO-HAS-B cells[54]. Therefore, the decreased expression of c-Myc with FoxO1 KD in ISO-HAS-B cells may be due to complexity in the aPKC/FoxO/miR-34c/c-Myc signaling axis under these conditions and will be examined further in the future.

Although angiosarcoma is an EC-derived tumor and over-expresses both VEGF-A and VEGF receptors (VEGFRs), the effect of anti-VEGF reagents including Bevacizumab and Sorafenib are limited[55,56]. While a recent study has shown that chemoradiotherapy with taxane may be an effective treatment for angiosarcoma, the prognosis for these patients is very poor[57]. Here we were able to show the association between aPKC expression level and the presence of pSer218-FoxO1 staining and patient survival and that aPKC inhibitors can affect angiosarcoma proliferation in vitro, hence, our results present a potential diagnostic strategy and may open a therapeutic opportunities for angiosarcoma.

aPKCλ expression is a prognostic marker in many malignant cancers including ovarian, breast, and lung with increased expression correlating with poor patient outcomes[13–16]. Conversely, FoxO transcription factors are tumor suppressors[58]. Nuclear localized FoxO transcription factors are known to induce cell cycle arrest, DNA repair, and apoptosis[58]. Supporting their role as tumor suppressors, degradation of FoxOs is observed in oncogenesis in human patients[59] and constitutive nuclear exclusion of FoxOs is assumed to promote tumor formation[58]. Indeed, constitutive activation of the PI3K/Akt pathway caused by mutation in Ras small GTPase, PTEN, a negative regulator of PI3K, and PI3K/Akt itself is observed in many types of cancer[60]. Consequently, FoxO nuclear exclusion as a result of PI3K/Akt mutation has been observed in ovarian cancer, breast cancer, and glioblastoma[22,61,62]. Additionally, many conventional anti-cancer therapies, including tyrosine kinase inhibitors, indirectly target FoxO activation[63]. In fact, treatment with tyrosine kinase inhibitors has been shown to induce nuclear translocation of FoxOs in a murine model of chronic myeloid leukemia, resulting in the induction of apoptosis[64]. Despite this established role as tumor suppressors, several drug resistant tumor cell lines derived from breast cancer and leukemia patients display FoxO overexpression and nuclear localization[65–67]. Moreover, elevated FoxO1 expression and its nuclear localization were observed in taxane-induced drug resistance in ovarian cancer patients[68]. Thus, nuclear localized FoxO in specific types of drug resistant tumors could be selectively inactivated by aPKCλ phosphorylation and therefore targeted therapy for the FoxO/aPKCλ signaling axis identified here may be useful.

## Methods

**Human patient samples.** Samples were obtained after ethical approval by the institutional Review Board at MD Anderson Cancer Center, the Institutional Review Board at Hokkaido University Hospital, or the ethics committee of Okayama University Graduate School of Medicine, Dentistry and Pharmaceutical Sciences, and Okayama University Hospital respectively.

**Mice.** Experiments involving animals were conducted in accordance with institutional guidelines and laws, and following the protocols approved by the local animal ethics committees and authorities (Regierungspraesidium Darmstadt). Pdgfb-iCre transgenic mice were bred into a background of animals containing a loxP-flanked Prkci[17] or loxP-flanked Foxo[8] or both. Cre activity in male and female neonatal mice was induced by intra-peritoneal injection of tamoxifen (1 mg/ml in ethanol/peanut oil, Sigma, T5648) of 50 μl from P1–P3[17]. The phenotype of the mutant mice was analyzed at P6 and tamoxifen injected Cre− littermates were used as controls. For mosaic deletion analysis, Prkci^iΔEC mice were crossed with B6.129 × 1-Gt (ROSA)26Sor^tm1(EYFP)cos]J fluorescent reporter mice. A single dose of tamoxifen (0.1 mg/mL) was injected into male and female neonatal mice on P1.

**Cell lines and primary cultures.** Pooled human umbilical vein endothelial cells (HUVECs) were purchased from Pelobiotech (Frankfurt, Germany) and cultured in EGM-2 medium (Lonza; Basel, Switzerland). HUVECs were used between passage 1 and 5. HEK293 cells were obtained from the American Type Culture Collection (LGC Standards, Molsheim Cedex, France) and cultured in DMEM (Sigma) with 10% fetal bovine serum (FBS; Biochrom GmbH, Berlin, Germany), 2 mM L-glutamine (Gibco, Invitrogen, Life Technologies; Darmstadt, Germany), 100 IUml−1 penicillin, and 100 mg/ml streptomycin. Angiosarcoma cell lines ISO-HAS-B and AS-M were provided by Dr. Mikio Masuzawa and Dr. Ronald E. Unger and were cultured in Endothelial cell growth medium MV (PromoCell; Heidelberg, Germany).

**Gene knockdown and overexpression strategies.** Oligonucleotide siRNA duplexes were purchased from GE Healthcare Dharmacon inc. SMART-pool ON-TARGET plus FOXO1 (see sequences Supplementary Table 2) and ON-TARGET plus PRKCI siRNA or scrambled control were transfected into HUVECs using Oligofectamine (Invitrogen) according to the manufacturer's instructions. Cells were analyzed 48 h post transfection. For overexpression of proteins in HUVECs, the indicated plasmids were transfected into the cells with Jet-PEI-HUVEC (Polyplus; Illkirch, France) according to the manufacture's instructions and analyzed 24 h post transfection.

Knockdown of aPKCλ (PRKCI) or FOXO1 was achieved using the same oligonucleotide siRNA duplexes in ISO-HAS-B cells after transfection with Lipofectamine-2000 (Invitrogen) according to the manufacturer's instructions. Cells were analyzed 48 h post transfection for aPKCλ KD. For aPKC inhibitor treatment after FoxO1 KD inhibitors were added 24-32 h post transfection and analyzed 48 h post transfection. MiR-34c inhibition in ISO-HAS-B cells was achieved by adding anti-miR-34c-5p or anti-miR scrambled control miRCURY LNA Pwr inhibitor (Exiqon) directly to the cell culture medium followed by analysis after 48 h of incubation. Overexpression was also carried out using Lipofectamine-2000 and the indicated plasmids, with analysis 24 h post transfection.

For overexpression of exogenous protein in HEK-293 cells, cells were transfected with PEI-MAX (Polysciences; Hirschberg an der Bergstrasse, Germany) according to manufacturer's instructions.

**Retina staining.** For retina staining, eyes were dissected and fixed in 4% PFA for 30 min on ice. Retinas were further dissected and permeabilized and blocked in 1% BSA (Sigma, A4378) and 0.3% Triton X-100 overnight at 4 °C with gentle rocking. Next, they were washed three times in Pblec buffer (1 mM CaCl2, 1 mM MgCl2, 1 mM MnCl2, and 1% Triton X-100 in PBS) and incubated with biotinylated isolectin B4 (Vector, B-1205, Griffonia simlicifolia lectin I, 1:50) and primary antibodies, overnight at 4 °C with gentle rocking: anti-c-Myc (Millipore 06-340, 1:100), anti-ERG1/2/3 (Abcam 92513, 1:500), anti-FoxO1 (Cell Signaling 2880, 1:100), anti-Claudin 5 (Zymed 341600, 1:100), anti-Collagen IV (Biorad 2150-1470, 1:400), anti-phospho-aPKC (Abcam ab59412, 1:100) anti-VE-cadherin (eBioscience 555289, 1:200), anti-CD31 (BD biosciences 553370, 1:200), anti-podocalyxin (R&D Systems AF1556, 1:200), anti-JAMA[55], or anti-pSer218-FoxO1 (1:100). Polyclonal anti-pS218 FoxO1 antibody was produced against Ile213-Arg-His-Asn-Leu-phospho Ser218-Leu-His-Ser-Lys-Phe223 by Eurogentec (Köln Germany). Retinas were washed five times with 0.5% BSA and 0.15% Triton X-100 and incubated with Alexa-Fluor-coupled streptavidin (Invitrogen, 1:100) and the corresponding Alexa-Fluor-coupled secondary antibody (Invitrogen, 1:500) in blocking buffer for 2 h at room temperature. Nuclei were stained with DAPI (Sigma-Aldrich 1:1000) and flat-mounted using Fluromount-G (SouthernBiotech, 0100-01). All images were captured using a Leica SP-8 confocal microscope.

**Phosphorylation assay.** The reaction was carried out in a 50 μl assay mixture containing 20 mM Tris-HCl (pH 7.4), 5 mM MgCl2, 1 mM EGTA, 1 mM DTT, 0.2 μM recombinant aPKCλ (Millipore), 10 μM ATP, and 1 μM purified GST-FoxO1

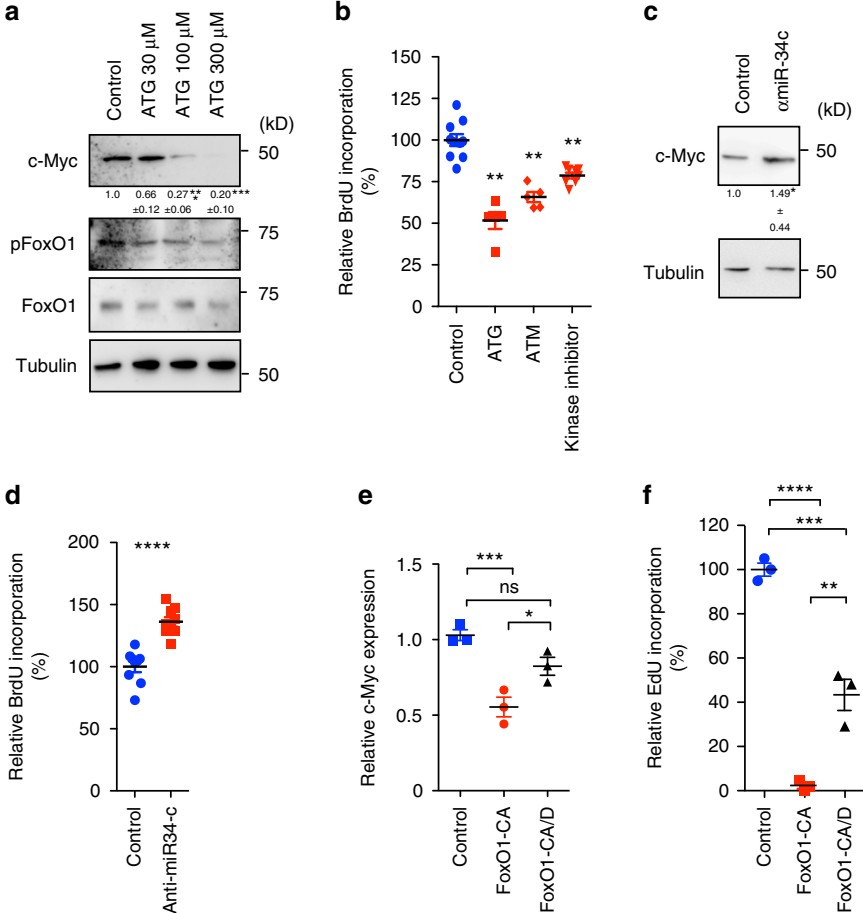

**Fig. 7** aPKC and miR-34c control angiosarcoma proliferation. **a** Western blot analysis of c-Myc, pSer218-FoxO1, and total FoxO1 in ISO-HAS-B angiosarcoma cells after treatment with aPKC inhibitor ATG for 24 h. Densitometric quantifications are shown below the lanes mean ±S.E.M.; one-way ANOVA with Bonferroni post-hoc analyses; ***$p < 0.001$; ($n = 3$). **b** Relative BrdU incorporation in ISO-HAS-B cells after treatment with aPKC inhibitors ATG, ATM, or aPKC kinase inhibitor after 24 h. Data represent mean ± S.E.M. two-tailed unpaired $t$-test **$p < 0.01$ ($n = 6$) **c** Western blot analysis of c-Myc expression in ISO-HAS-B cells after treatment with anti-miR-34c or scrambled control. Densitometric quantifications are shown below the lanes mean $+/-$ S.E.M.; two-tailed unpaired $t$-test; ($n = 3$); *$p < 0.05$. **d** Relative BrdU incorporation in ISO-HAS-B cells after treatment with anti-miR-34c or scrambled control. Data represent mean ± S.E.M. two-tailed unpaired $t$-test ****$p < 0.0001$ ($n = 6$). **e** Relative c-Myc expression measured by immunofluorescence in FoxO1-CA or FoxO1-CA/D expressing cells compared to non-transfected neighboring cells; Data represent mean ± S.E.M.; one-way ANOVA with Bonferroni post-hoc analysis; ns $p > 0.05$ vs. control; *$p < 0.05$ CA vs. CA/D; ***$p < 0.01$ vs. control. **f** Relative EdU incorporation in FoxO1-CA or FoxO1-CA/D expressing cells compared to non-transfected neighboring cells; Data represent mean ± S.E.M. one-way ANOVA with Bonferroni post-hoc analysis; **$p < 0.01$ CA vs. CA/D; ***$p < 0.01$ vs. control, ****$p < 0.001$ vs. control

for 60 min at 30 °C. The reaction was stopped by the addition of SDS sample buffer and the products were subjected to SDS-PAGE, followed by a CBB staining. Radiolabeled bands were visualized with an image analyzer (Fuji BAS 4000).

**Ultra high pressure liquid chromatography and mass spectrometry**. Extracted peptides from in-gel digestion were desalted, concentrated, and re-suspended in acidified water (0.1 % formic acid). Peptides were separated on nanoLC 1000 (Proxeon, now Thermo Fisher) using a binary buffer system of A (0.1 % (v/v) formic acid in H$_2$O) and B (0.1 % (v/v) formic acid in 80% Acetonitrile)[69]. We applied a linear gradient from 7 to 35% B for 35 min followed by 95% B for 15 min and re-equilibration to 5% B within 5 min (held for further 5 min at 5 %) on a 50 cm column (75 μm ID) in-house packed with C18 1.8 μm diameter resin (Dr. Maisch, Germany). To control the column temperature we used an in-house made column oven at 40 °C. The UHPLC was coupled via a nano-electrospray ionization source (Thermo Scientific) to the quadrupole based mass spectrometer QExactive (Thermo Scientific)[69]. MS spectra were acquired using 3e6 as an AGC target at a resolution of 70,000 (200 $m/z$) in a mass range of 350–1650 $m/z$. A maximum injection time of 60 ms was used for ion accumulation. In a data dependent mode MS/MS events were acquired for the ten most abundant peaks (Top10 method) in the high mass accuracy Orbitrap after HCD (Higher energy C-Trap Dissociation) fragmentation at 25 collision energy in a 100–1650 $m/z$ mass range. The resolution was set to 17,500 at 200 $m/z$ combined with a maximum injection time of 60 ms.

For data analysis all 27 raw files were processed using MaxQuant (1.3.7.4) and the implemented Andromeda search engine[70]. For protein assignment ESI-MS/MS

fragmentation spectra were correlated with the Uniprot human reference proteome database including a list of common contaminants. Searches were performed with Trypsin specificity allowing two missed cleavages and a mass tolerance of 4.5 ppm and 7 ppm for MS/MS first and main searches, respectively. Carbamidomethyl at cysteine residues was set as a fixed modification and phosphorylation (STY), oxidation at methionine, and acetylation at the N-terminus were defined as variable modifications. The minimal peptide length was set to 7 amino acids and the false discovery rate for proteins and peptides to 1%. The FDR was applied separately for modifications using the implemented decoy approach in revert mode. Identified phosphorylation sites with specific scores and sequence windows are shown (Supplementary Fig. 3A). Fold changes were calculated based on raw intensities.

**Protein purification**. The gene encoding full-length human FoxO1, full-length human S218A FoxO1, wild-type human FoxO1 DNA-binding domain amino acids 151–266, and S218D human FoxO1 DNA-binding domain amino acids 151–266 proteins were sub-cloned into pDEST 15 (Thermo Scientific), produced in Rosetta (DE3) *Escherichia coli* and purified on gluathione-Sepharose 4B beads (GE Healthcare).

**Phosphorylation assay in ECs**. For transient overexpression of pCAGGS-myc-aPKCλ and pcDNA3-FLAG-FoxO1[21] or empty vector control in HUVECs, DNA was transfected into a 6-well plate using jetPEI (Polyplus; Illkirch, France) according to the manufacturers instructions. After 24 h of post transfection, the

cells were washed with PBS and harvested with SDS sample buffer. Samples were subjected to western blot analysis with anti-pSer218-FoxO1, anti-Myc (No. sc-40; Santa Cruz Biotechnologies; Heidelberg, Germany), and anti-Flag-M2 antibodies (No. F1804; Sigma)

**Nuclear fractionation.** HUVECs were lysed 48 h post siRNA KD or 24 h after transfection of pCAGGS-myc-aPKCλ and pcDNA3-FLAG-FoxO1 in a hypotonic buffer (10 mM HEPES, 1.5 mM MgCl$_2$, 10 mM KCl, 0.5 mM DTT, and 0.05 % NP-40) and then centrifuged for 10 mins at 16,000× $g$ at 4 °C. The supernatant representing the cytoplasmic fraction was collected and the pellet was washed twice with the hypotonic buffer. Pellets were then re-suspended and lysed in a high salt buffer (450 mM NaCl, 50 mM Tris pH 7.5, 2 mM DTT, 1% NP-40, protease inhibitor (Sigma P2714), and phosphatase inhibitor (Calbiochem 524629; Darmstadt, Germany) and incubated for 10 min with rotation at 4 °C. Following centrifugation at 16,000× $g$ for 15 min the supernatant was collected and snap frozen at −80 °C. For normalization of loading, a BCA protein assay (Pierce; Schwerte, Germany) was carried out and 16 µg of protein was loaded onto the gel in 6xSDS loading buffer and western blot analysis was carried out according to standard laboratory practices. Western blot analysis of fractionated samples was carried out with the following antibodies: anti-FoxO1 (Cell Signaling Technology; No. 2880), anti-Lamin A/C (Santa Cruz Biotechnologies; No. sc-6215), anti-Lamin B1 (Santa-Cruz Biotechnologies; No. sc-6216), anti-α-tubulin (Cell Signaling Technology; No. 2125), anti-aPKC (BD Biosciences; No. 610207), or phospho-aPKC (Abcam; ab59412).

**DNA-binding Assay.** EMSAs were performed with biotinylated DNA duplexes containing the DBE sequence strands 5′–CAAAATGTAAACAAGA-3′ and 5′-TCTTGTTTACATTTTG-3′ (Sigma). Complementary strands of duplex DNA were annealed in 20 mM HEPES (pH 7.5) and 50 mM NaCl buffer by heating to 80 °C for 10 min and then slowly cooling to room temperature for 3 h. WT-FoxO1 or S218D-FoxO1 DNA-binding domain at a final concentration of 1 µM were mixed with 1 nM of DNA and incubated at RT for 30 min in binding buffer containing 20 mM Tris (pH 8.0), 50 mM KCl, 5% glycerol, 2 mM DTT, 0.2 mM EDTA, 2 mM MgCl$_2$, 0.1 mg/mL BSA, and 1 ng/mL poly dI-dC. The mixture was then loaded onto a 6% DNA retardation gel in 0.5X TBE and run at 100 V for 1 h at 4 °C. The gel was then blotted to a Biodyne B membrane (Invitrogen) at 380 mA for 1 h in 0.5X TBE at 4 °C. The DNA was cross-linked to the membrane using a Stratagene cross-linker and developed using the chemiluminescent nucleic acid detection kit (Pierce) according to manufacturers instructions.

**Chromatin-immunoprecipitation.** HUVEC were transfected with pcDNA3-FLAG-FoxO1-3A[21] constitutively active mutant, pcDNA3-FLAG-FoxO1-3A-S218D mutant or empty vector control and 24 h after transfection 10 × 10$^6$ cells were fixed for 30 seconds with 3% formaldehyde and cross-linking was stopped with 2.5 M glycine. Cells were lysed in lysis buffer (5 mM PIPES; 85 mM KCl; 0.5% NP-40; Protease inhibitor (Sigma)) and nuclei were isolated by passing lysate through a 27 g needle. Nuclei were then sheared using a Bioruptor sonicator for 2 × 10 cycles (30 s on/off) in RIPA buffer to a size of 200–300 bp. Sheared chromatin was then incubated with FLAG-M2-conjugated Protein-G Dynabeads (Sigma and Invitrogen). Cross-links were reversed by overnight incubation at 65 °C followed by Proteinase-K treatment at 37 °C for 1 h and cleaned up with the Qiaquick PCR clean up kit. Precipitated chromatin and inputs were amplified by quantitative PCR.

**Luciferase assays.** HEK293 cells were transfected with pGL3-6xDBE and pGL4-hRluc and 8 h post transfection cells were washed with PBS and medium was replaced or changed to DMEM + 0.1% BSA with or without 5 µM aPKC kinase inhibitor (539624, Calbiochem) and incubated overnight. For experiments using the CTDSP2/CTDSP2-mut luciferase constructs, luciferase vectors[20] and pGL4-hRluc were transfected with vector controls and pCAGGS-myc-aPKCλ and wild-type pcDNA3-FLAG-FoxO1. pGL3-6xDBE and pGL4-hRluc were also co-transfected with either pcDNA3-FLAG-FoxO1-3A mutant or pcDNA3-FLAG-FoxO1-3A-S218D mutant. All experiments were collected 24 h post transfection and luciferase and *Renilla* expression were measured using the Dual-Luciferase Reporter Assay (Promega) using a Mithras LB940 plate reader (Berthold Technologies), according to the manufacturers instructions and luciferase luminescence was normalized for input to *Renilla* luminescence.

**RNA extraction and qPCR analysis and western blotting.** For analysis of messenger RNA expression levels total RNA was isolated from HUVEC using the Quick RNA-Mini kit (Zymo Research; Freiburg, Germany) and from control or mutant retinas using the RNeasy Mini Kit (Qiagen). A volume of 0.5–1 µg of RNA was used per reaction to generate cDNA with the Superscript VILO cDNA synthesis kit (Invitrogen; HUVEC) or iScript cDNA synthesis Kit (Bio-rad; retinas). qPCR was carried out using a StepOnePlus real time PCR machine using in-house designed primers or pre-made primer sets (Applied Biosystems, Calsbad, CA). Sequences of pre-made primers and in-house designed primers are shown in supplementary table 3. TaqMan gene expression assay for *PKM2* and *Cldn5* were used in combination with TaqMan Fast Gene Expression Master Mix (Applied

Biosciences) and expression was normalized to endogenous *GAPDH* also using the Taqman gene expression assay. Primer information is shown in supplementary table 4. All other targets were analyzed using SYBR-Green PCR Master Mix (Applied Biosystems). Gene expression assays carried out using SYBR-Green were normalized to endogenous expression of *B2M*.

Samples for western blotting were collected into 1xSDS-sample buffer and subjected to immunoblotting with anti-c-Myc (Santa Cruz Biotechnology; sc-40 or Millipore; 06-340), anti-FLAG (Sigma; 8592); anti-aPKC (BD Biosciences; 610207), or in-house prepared pSer218-FoxO1 and normalized to anti-α-tubulin (Cell Signaling Technology; 2125). See example uncropped blots in the Supplementary figure 11.

For western blot analysis from the formalin-fixed paraffin embedded angioscaroma patient sample, five 10 µm sections were freshly cut and placed in a tube. The sections were first deparaffinized with xylene and then rehydrated in a graded alcohol series and allowed to air dry. Subsequently, a 300 mM Tris-HCl pH 8.0 and 2% SDS solution was added to the tube and incubated for 5 min at room temperature. The samples were then sonicated in a Bioruptor ultrasonicator for 2 × 30 sec and boiled at 100 °C for 20 min. The samples were further incubated at 80 °C for 2 h and then centrifuged for 20 min at 16,000× $g$. 6x SDS sample buffer was then added to the supernatant and samples were then boiled again for 10 min and loaded onto the gel.

**miRNA extraction and qPCR analysis.** Total small RNAs from HUVEC, ISO-HAS-B, and mouse retina were collected using the mirVANA miRNA isolation kit (Ambion) or miRNeasy mini kit (Qiagen). For isolation of miRNA after aPKC inhibitor treatment, HUVEC or ISO-HAS-B were incubated overnight in EBM + 0.1% BSA with or without 5 µM aPKC kinase inhibitor (539624, Calbiochem) or sodium aurothioglucose (ATG; Sigma). The reverse transcription reaction was carried out using 10 ng of total small RNAs and the TaqMan advanced miRNA cDNA synthesis kit (Applied Biosystems).

miR-34c-5p and miR-145-5p expression were analyzed with TaqMan Advance miRNA assays and expression was normalized to miR-186-5p control. Sequences of primers are shown in Supplementary Table 4.

**In vivo anti-miR treatment.** *Prkci*$^{iΔEC}$ pups were injected intra-abdominally on P1 and P3 with either anti-miR-34c-5p or anti-miR scrambled control miRCURY LNA Pwr inhibitor (Exiqon) at a final dose of 10 mg/Kg. Phenotype was analyzed at P6 and tamoxifen injected *Cre*$^−$ littermates were used as controls.

**In vitro cell proliferation assay.** HUVEC, ISO-HAS-B, and AS-M were re-seeded onto triplicate wells in a 96 well plate at a density of 5000 or 10,000 cells per well. Cells were incubated overnight or for 24 h with or without aPKC kinase inhibitor, sodium aurothioglucose (ATG, Sigma), or sodium aurothiomalate hydrate (ATM, Sigma) and then BrdU was added to the culture medium and the cells were incubated for a further 2 h. BrdU incorporation was then detected using the Cell proliferation ELISA colorimetric kit (Roche; 11647229001).

Alternatively, ISO-HAS-B cells were grown on gelatin-covered coverslips and subsequently transfected with either scrambled control or FoxO1 siRNA or FoxO1-CA or FoxO1-CA/D. Twenty four hours post FoxO1, siRNA transfection medium was changed to medium with or without aPKC kinase inhibitor and the cells were incubated for a further 24 h. Two hours before the end of the incubation period (24 h for FoxO1 mutant overexpression and 48 h for FoxO1 siRNA), EdU was added to the cell culture medium according to the manufacturer's instructions and EdU incorporation was examined using the Click-It$^{TM}$ EdU Alexa Fluor 488 Imaging Kit (Invitrogen). Nuclei were visualized with Hoescht 33342. To identify cells expressing FoxO1-CA/FoxO1-CA/D, after the completion of the Click-It$^{TM}$ reaction cells were stained with anti-FLAG-M2 (1:400; Sigma) overnight at 4 °C, followed by washing and incubation with Alexafluor donkey-anti-mouse 555 and finally Hoescht 33342. The proportion of FLAG positive EdU positive cells was compared to the proportion of FLAG negative EdU positive neighboring cells.

**Immunofluorescent staining of HUVEC and ISO-HAS-B cells.** HUVEC and ISO-HAS-B cells were cultured on gelatin coated coverslips and were either left untreated or transfected for ectopic expression of FoxO1 mutants. Cells were fixed for 10 min with 4% paraformaldehyde, washed with PBS and then permeabilized with 0.5% Triton-100 × -PBS for 10 min. Subsequently cells were blocked with 5% FCS 0.01% Tween-PBS for 30 min followed by incubation with anti-aPKC (1:100; Beckton Dickison), anti-VE-cadherin (1:100; R&D Systems); anti-c-Myc (1:100; Millipore); and anti-FLAG-M2 (1:400; Sigma) overnight at 4°C. Coverslips were then washed and incubated with the appropriate secondary antibodies (Alexafluor fluorescently conjugated; 1:400; Invitrogen) and Hoescht 33342 or DAPI, and mounted with Vectashield.

**Angiosarcoma and pyogenic granuloma tissue sample immunostaining.** Formaldehyde-fixed paraffin block patient tissue samples were cut into 5-7 µM sections then deparaffinized in xylene and rehydrated in a graded ethanol series. This was followed by antigen retrieval in sodium citrate buffer by boiling the slides for 15 min. The tissue was then blocked in 5% FCS + 0.1% Tween-20 for 30 min at room temperature followed by incubation with primary antibodies overnight at 4

degrees (anti-FoxO1 1:100; Abcam 39670; anti-c-Myc 1:50; Millipore 06-340; anti-aPKC 1:25; Becton Dickinson; 610207; anti-pSer218-FoxO 1:100; in-house pre-paration; anti-VE-Cadherin 1:100; R&D Systems AF938; anti-Ki67 1:100; Abcam 15580). Tissue was then washed and secondary antibodies were applied. For rabbit primary antibodies (FoxO1, c-Myc, pSer218-FoxO), tissue was first incubated for 1 h at room temperature with biotin-conjugated donkey-anti-rabbit (Jackson Immunoresearch 1:200; 711-065-152), then washed and incubated with donkey-anti goat and streptavidin conjugated secondary antibodies (Invitrogen 1:200) again for 1 h. For anti-aPKC mouse antibody double staining with anti-VE-Cad-herin, slides were first incubated with donkey-anti goat secondary (Invitrogen 1:200) for 1 h at room temperature, followed by goat-anti mouse secondary (Invitrogen 1:200) for an additional hour. All slides were then incubated with Hoechst 33342. Ki67 index was calculated as the proportion of Ki67 positive VE-Cadherin positive cells in a tissue section and samples were classified as Ki67 high when the Ki67 index was above the average calculated for the samples and low when below average. Staining intensity and localization were evaluated by three investigators independently.

**c-Myc fluorescent in-situ hybridization**. The presence of c-Myc gene amplifi-cation was examined in c-Myc overexpressing patient samples using the kit Zyto*light* SPEC CMYC/CEN 8 dual color probe (Zytovision; Bremerhaven, Ger-many) and Zyto*light* FISH-Tissue implementation Kit (Zytovision) according to manufacturer's instructions.

**Quantification and Image processing**. Data are based on at least three inde-pendent experiments or three mutant and control animals for each stage and result shown. The data are presented as mean ±S.E.M. and exact statistical analyses carried out are detailed in the figure legends. All statistical analyses were carried out using Prism 5 software. A $p < 0.05$ was considered significant.

Volocity (Improvision) was used for quantitative image analysis. Volocity, Photoshop CS, and Illustrator CS (Adobe) software were used for image processing and in compliance with general guidelines for image processing.

The authors can confirm that all relevant data are included in the paper or its supplementary information files.

## Data availability

The remaining data are available within the article and its supplementary infor-mation files and from the corresponding author on reasonable request. The mass spectrometry proteomics data have been deposited in the ProteomeXchange Consortium via the PRIDE partner repository under the accession code PXD011595.

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

## Acknowledgements

We thank Dr. Ronald Unger and Dr. Vlad Cojocaru for reagents and discussions, and Dr. Markus Krueger for mass spectrometric analysis. Funding for this project was provided by the Excellence Cluster Cardio-Pulmonary System, the German Research Foundation, Deutsche Forschungsgemeinschaft (NA 1195/1-1). M.R. is supported by the Alexander von Humboldt Foundation.

## Author contribution

M.R. and M.N. designed the study. M.P., T.Hirose. and S.O., provided the mutant mouse lines. M.Looso. performed bioinformatics analysis. M.M., K.E., and I.F. contributed experimental reagents and insights. S.G., P.P.A., T.P., O.Y., T.Y., and H.U prepared patient-derived tissue samples. All other experiments were performed by M.R., A.N., T. Hikita., F.M., T.K., A.P., M.Li, T.S. and M.N. M.R. and M.N. wrote the manuscript.

## Additional information

**Competing interests:** The authors declare that they have no competing interests.

