## [Peer Review file · Nature Communications]

Reviewers' comments:

Reviewer #1 (Remarks to the Author):

In this manuscript, the authors convincingly demonstrate a novel mechanism by which endothelial cell proliferation is regulated via FOXO1. Additionally, they have uncovered a novel signaling axis whereby aPKC-mediated phosphorylation of FOXO1 greatly impairs FOXO1 gene regulatory activity independently of PI3K/Akt-mediated phosphorylation and related changes in subcellular localization. This last finding may prove to be of great fundamental importance to the FOXO field as it broadens the repertoire of signaling pathways now known to regulate FOXO nuclear function. However, there are some points that should be addressed in the revision to strengthen this manuscript.

1) Throughout the figures western blots for total and pSer218 FOXO1 are sometimes absent or of low quality where they are critical to assess the phosphorylation state of FOXO1 on the novel Ser218 site. In fig. 2c the total FOXO1 western seems to have some background signal, fig. 7a FOXO1 total is missing, fig. S8b adding total and pSer218 FOXO1 would benefit the cell line comparison and in fig. S8e FOXO1 total is missing and a very low-quality blot of pSer218 is included.

2) In fig. 2c it seems that FOXO1 total levels are significantly increased by both aPKC overexpression as well as aPKC pharmacological inhibition. In fig. S4a aPKC knockdown seems to decrease the FOXO1 cytosolic pool significantly; especially if the cytosolic control is as underloaded as the alpha-tubulin loading control suggests. This raises the possibility that changes in pSer218 levels observed elsewhere under aPKC manipulation may be due at least in part to changes in FOXO1 protein levels. The authors should perform cycloheximide pulse-chase experiments of FOXO1 total levels under both baseline and aPKC inhibited conditions to gauge FOXO1 stability which may be impacted by Ser218 phosphorylation. These experiments might also be considered with the WT and S218D FOXO1 constructs. These experiments would also be relevant in appreciating the degree to which pSer218 impacts FOXO1 nuclear activity via purely DNA binding or additional mechanisms.

3) The functional demonstration of the aPKC/FOXO1/mir-34c/c-Myc axis in the ISO-HAS-B cell line is somewhat incomplete. Crucially, FOXO1 should be knocked down in ISO-HAS-B to see if this can revert the mir-34c upregulation, c-Myc suppression and growth inhibition observed when aPKC is inhibited either pharmacologically or genetically. An additional experiment would be to overexpress FOXO1-CA and FOXO1-CA S218D in ISO-HAS-B and assess growth arrest as well as mir-34c and c-Myc regulation; there should be significant differences between these constructs here.

Reviewer #2 (Remarks to the Author):

In this article, Ridell et al. dress the long-known conundrum that many tumors show nuclear overexpression of FoxO1, an established tumor suppressor. The authors show here a) that aPKC λ directly controls FoxO1 activity via phosphorylation in a manner independent of its subcellular localization and b) that this pathway modulates c-MYC expression during physiological angiogenesis and in angiosarcomas through inhibition of a negative regulator of c-MYC, miR-34c.

Their findings are novel and explain previous seemingly paradoxical observations of co-expression of FOXO1 and c-MYC and open new therapeutic perspectives in tumors with activation of the aPKC λ /FoxO1 pathway through the application of PKC λ inhibitors.

Specific comments: while the part on angiogenesis is very strong, their findings in angiosarcomas are of a more correlative nature (immunohistochemical co-expression of aPKC λ , FoxO1, and c-

MYC). Gene amplification of c-MYC has been observed especially in a high proportion of radiation-induced angiosarcomas (Manner et al. 2010, Guo et al. 2011), which might also explain overexpression of c-MYC. The authors did not provide data on whether some of their cases were radiation-induced. Was the MYC-amplification status checked?

The expression level of miR-34c in correlation with FoxO1 and c-MYC levels should be checked in the clinical samples.

The expression level of aPKC λ , FoxO1, and c-MYC should be correlated with proliferation rate (e.g. ki67 index) in the angiosarcoma samples

Reviewer #3 (Remarks to the Author):

In this interesting manuscript the authors identify as a novel target for atypical PKC lambda/iota (aPKC) the transcription factor Foxo1. The focus is on endothelial cells, and the retinal vasculature is used as a model tissue. Although aPKC negatively regulates VEGF, knockout of aPKC in endothelial cells reduces proliferation and vascular branching. This correlated with reduced c-Myc expression. Foxo1 KO mice show increased vascular coverage, which was not altered in double Foxo1/aPKC KOs, and c-Myc levels were high – suggesting that aPKC controls c-Myc expression through Foxo1.

The authors show convincingly that aPKC phosphorylates Foxo1 on Ser218, which is in the DNA binding domain, and phosphorylation, or mutation of S218 to aspartate blocks binding to Foxo1 sites and the activation of transcription. However, there is no effect on Foxo1 localization. Next, the authors demonstrate that Foxo1 regulates the micro-RNA miR-34c, which targets c-Myc mRNA for degradation. This miRNA is a known regulator of c-Myc, and is known to be induced by Foxo1, so this part of the story is not particularly novel. However, they use a miR inhibitor of miR-34c to show that this mechanism functions in vivo. Finally, they show that this aPKC/Foxo1/c-Myc signaling pathway plays a role in angiosarcoma, and demonstrate high levels of P-S218 Foxo1 in a large fraction of angiosarcoma samples, which correlated with poor survival probability. Overall, this is a very thorough study that links aPKC to transcriptional regulation of c-Myc and endothelial cell proliferation. The authors have previously shown that active, phosphorylated aPKC is reduced at the angiogenic front. However, they do not investigate the underlying mechanism regulating the spatial activity of aPKC and it is unclear what might cause this. This gap in the regulatory pathway should at the least be discussed.

There are also a few technical points that the authors need to address:

1. In the abstract they refer to aPKC as an oncogene, but provide no support for this claim in the introduction, and although it is often over-expressed in tumors and mislocalized the data supporting a function as a bona fide oncogene is at best circumstantial (the one cited paper in Cancer Res claiming aPKC is an oncogene in NSCLC is just correlative).
2. Although in this study the aPKC pseudosubstrate kinase inhibitor seems to be working through aPKC, the authors should acknowledge that it is quite nonspecific. Indeed, its role in inhibiting a splice variant of aPKC (PKM) in the brain to block memory formation has been discredited.
3. In Figure 2f the left and right panels do not seem to correspond – there is abundant red staining in the right but not the left panel. Perhaps the colors were switched? Also, in Figure 2e the staining is not very convincing (Right panel)
4. Figure 5 – why is the c-Myc staining not localized to the nuclei here? Is most of the staining nonspecific?
5. Figure 6 – what are the values of the x-axis? (months? Weeks?)
6. Figure 7 b, d – the y axes are not labeled.
7. Figure S6a – the podocalyxin looks to be cytosolic, not apical; and why is the collagen IV staining also apical, especially in the aPKC KO image?

We would like to thank all of the reviewers for their high evaluation of the paper and their constructive feedback working towards improving our manuscript. As you will see, we have extensively revised the manuscript by adding a large amount of new data. While a detailed point-by-point response to all comments is given further below, here is a brief summary of the most important additions.

1. The indicated low-quality western blot panels were replaced with better blots to avoid any confusion and misunderstanding.
2. To further demonstrate the functional importance of the aPKC/FoxO1/miR34-c/c-Myc signaling axis in ISO-HAS-B cells, we examined the effect of over expression of FoxO1-CA and FoxO1-CA/D on c-Myc expression and cell proliferation and loss of function of FoxO1 combined with aPKC inhibitor treatment in these cells.
3. Given the fact that gene amplification of c-MYC is observed, especially in a high proportion of radiation-induced angiosarcomas, we checked the c-Myc amplification status of our patient samples. Although none of our samples are cases of radiation-induced angiosarcoma, we found c-Myc amplification in one patient sample. We excluded this patient from the survival curve analysis and confirmed a strong correlation of FoxO1 phosphorylation by aPKC as well as aPKC expression with patient prognosis.
4. To strengthen our conclusion that phosphorylation of FoxO1 by aPKC is a key factor associated with proliferation of transformed cells in angiosarcoma, the expression level of aPKC, FoxO1 and c-Myc was correlated with proliferation rate in the angiosarcoma samples. A strong association between c-Myc and aPKC expression and the Ki67 index was established.

Please find a point-by-point response below:

Reviewer #1:

1) *Throughout the figures western blots for total and pSer218 FOXO1 are sometimes absent or of low quality where they are critical to assess the phosphorylation state of FOXO1 on the novel Ser218 site. In fig. 2c the total FOXO1 western seems to have some background signal, fig. 7a FOXO1 total is missing, fig. S8b adding total and pSer218 FOXO1 would benefit the cell line comparison and in fig. S8e FOXO1 total is missing and a very low-quality blot of pSer218 is included.*

Thank you for the comment to improve the quality of our manuscript. Accordingly, all of the western blots mentioned above have now been replaced with higher quality examples and total Foxo1 has been included where requested (Fig. 7a, Supplementary Fig. 8b,e).

2) *In fig. 2c it seems that FOXO1 total levels are significantly increased by both aPKC overexpression as well as aPKC pharmacological inhibition. In fig. S4a aPKC knockdown seems to decrease the FOXO1 cytosolic pool significantly; especially if the cytosolic control is as underloaded as the alpha-tubulin loading control suggests. This raises the possibility that changes in pSer218 levels observed elsewhere under aPKC manipulation may be due at least in part to changes in FOXO1 protein levels. The authors should perform cycloheximide pulse-chase experiments of FOXO1 total levels under both baseline and aPKC inhibited conditions to gauge FOXO1 stability which may be impacted by Ser218 phosphorylation. These experiments might also be considered with the WT and S218D FOXO1 constructs. These experiments would also be relevant in appreciating the degree to which pSer218 impacts FOXO1 nuclear activity via purely DNA binding or additional mechanisms.*

We are sorry for causing confusion and appreciate how carefully and thoughtfully this reviewer examined each piece of data in the paper. Due to the inclusion of poor quality western blots on our part (original version Fig 2c. and FigS4a tubulin blot) it did appear that there was a possibility *in vitro* of aPKC knockdown or inhibition affecting the abundance of FoxO1. We have now replaced these poor quality western blots and it can be seen that in the experiments in question there is no change in FoxO1 abundance with siRNA KD or pharmacologic inhibition of aPKC. This data is also consistent with the phenotype we observe in the Prkci EC specific KO mice where no change in the expression level and localization of FoxO1 was observed.

3) *The functional demonstration of the aPKC/FOXO1/mir-34c/c-Myc axis in the ISO-HAS-B cell line is somewhat incomplete. Crucially, FOXO1 should be knocked down in ISO-HAS-B to see if this can revert the mir-34c upregulation, c-Myc suppression and growth inhibition observed when aPKC is inhibited either pharmacologically or genetically. An additional experiment would be to overexpress FOXO1-CA and FOXO1-CA S218D in ISO-HAS-B and assess growth arrest as well as mir-34c and c-Myc regulation; there should be significant differences between these constructs here.*

According to the reviewer's suggestion, we first examined the effect of overexpression of FoxO1-CA and FoxO1-CA S218D (FoxO1-CA/D) in ISO-HAS-B cells on c-Myc expression by immunostaining and cell proliferation with Edu incorporation due to low transfection efficiency in ISO-HAS-B cells. While ectopic expression of FoxO1-CA significantly reduced c-Myc expression and Edu

incorporation as expected, there was significant difference between the effect of FoxO1-CA/D and FoxO1-CA (Fig. 7e,f). We were, unfortunately, unable to carry out the requested miR-34c analysis with FoxO1 mutant overexpression due to very low transfection efficiency (~15%). When ISO-HAS-B cells are treated with aPKC inhibitor, we observed increased miR34-c expression, while this was not the case when aPKC inhibitor was added to FoxO1 KD cells. Consistently, aPKC inhibitor treatment resulted in reduced EdU incorporation and there was no additive effect of FoxO1 KD on EdU incorporation (Fig. S9c). These results strongly indicate that the effect of aPKC inhibition is in a manner dependent on FoxO1.

The puzzling results were that FoxO1 KD also induced miR34-c expression in ISO-HAS-B cells and reduced c-Myc expression. It has been shown that FoxO proteins interact with p53 regulating p53 activity and thereby shared target genes, including miR-34c (Miaguchi et al. Cell Biol International 2009; You et al. PNAS 2013; Rupp et al. Oncogene 2017, Sachdeva et al. PNAS 2008; He et al. Nature 2007). While p53 expression is normally low in healthy tissue, it's expression can be increased in many transformed cells and tumor tissues (Zeitzi et al. Am J. Pathol 1998; Italiano et al. Cancer 2012; Naka et al. Int J. Cancer 1997). Indeed, expression of p53 has been demonstrated in ISO-HAS-B cells (Masuzawa et al. Cancer Med 2012). Therefore, the decreased expression of c-Myc with loss of FoxO1 in ISO-HAS-B cells may be due to complexity due to p53 expression and dual regulation of miR34-c by FoxOs and p53 in specific conditions, such as in transformed cells. Interestingly, we could not observe an additive effect of FoxO1 KD and aPKC inhibition in ISO-HAS-B cells, suggesting aPKC may also be the upstream of p53. While further elucidation of the role of FoxO1 in ISO-HAS-B cells, particularly with regards to p53, is interesting, we have clearly shown the importance of FoxO phosphorylation by aPKC for endothelial proliferation during development and in angiosarcoma with many direct and indirect pieces of evidence. We have also, importantly, shown that FoxO1 phosphorylation by aPKC correlates with patient prognosis (Fig 6b). Thus, we believe the mechanism behind FoxO1 KD reducing c-Myc expression is outside the scope of this paper and it will be examined in the future.

We discussed this issue in the manuscript (page 16, line 326 to 337 and page 18 line 384 to 394).

Reviewer #2:

1) Specific comments: while the part on angiogenesis is very strong, their findings in angiosarcomas are of a more correlative nature (immunohistochemical co-expression of aPKC λ , FoxO1, and c-MYC). Gene amplification of c-MYC has been observed especially in a high proportion of radiation-induced angiosarcomas (Manner et al. 2010, Guo et al. 2011), which might also explain overexpression of c-MYC. The authors did not provide data on whether some of their cases were radiation-induced. Was the MYC-amplification status checked?

We apologize for leaving the information about the status of our angiosarcoma patients out of the original manuscript. All of our patients were cases of primary angiosarcoma and this information has been included in the manuscript (Page 13 Line 264). Additionally we carried out FISH for c-Myc on our patient samples with a high level of c-Myc expression. One patient (#11) was found to have high-level c-Myc amplification. This information is now included in the manuscript (Page 14 Line 272). To focus on the association between c-Myc expression and FoxO1 phosphorylation by aPKC, we excluded this patient from the survival curve analysis (Figure 6).

2) *The expression level of miR-34c in correlation with FoxO1 and c-MYC levels should be checked in the clinical samples.*

In order to attempt to address this comment we isolated miRNA from our patient samples. We were only able to isolate miRNA that was detectable by RT-PCR from 9 samples among our available samples. Values of the housekeeping microRNA miR-186 varied widely and miR-34c was detectable in a single pSer218-FoxO1 negative patient (Patient #38; Table 1 below).

Patient	miR34c (mean CT value)	miR186 (mean CT value)
18	Undetermined	25,52505684
7	Undetermined	32,26618958
35	Undetermined	31,27061462
34	Undetermined	30,26822281
33	Undetermined	28,46653748
12	Undetermined	31,54075623
37	Undetermined	23,74809265
9	Undetermined	27,72211838
38	35,85	23,68182373

Additionally, we carried out in-situ hybridization for miR-34c. Unfortunately, we could only see a U6 snRNA in-situ signal in limited samples as seen in the RT-PCR analysis above. U6 snRNA is a positive control used to determine the condition of RNA and suitability of samples for in-situ analysis (Jorgesen et. al Methods 2010). Given these facts, unfortunately, we concluded the condition of the RNA in our patient samples is very poor, and could not address this reviewer's comment. We will endeavor in the future to address this comment in full, however the samples collected for use in this paper are from multiple sites in several countries and have been collected over a period of >13 years, therefore, due to the rare nature of angiosarcoma we cannot fully address this comment in a reasonable time period.

3) *The expression level of aPKCλ, FoxO1, and c-MYC should be correlated with proliferation rate (e.g. ki67 index) in the angiosarcoma samples.*

We have now carried out this analysis and have included the data in the text of the paper (Page 14 line 287 to 292). As expected from the literature, we found a strong association between c-Myc and aPKC expression and the Ki67 index. While FoxO1 expression also showed clear tendency, we were unable to find a statistically significant association between FoxO1 expression level, though this is likely due to the low number of samples in the different categories of FoxO1 expression level.

Reviewer #3:

1) Overall, this is a very thorough study that links aPKC to transcriptional regulation of c-Myc and endothelial cell proliferation. The authors have previously shown that active, phosphorylated aPKC is reduced at the angiogenic front. However, they do not investigate the underlying mechanism regulating the spatial activity of aPKC and it is unclear what might cause this. This gap in the regulatory pathway should at the least be discussed.

We thank this reviewer for their very high evaluation of the manuscript. We have now included a section in the conclusions regarding the potential regulation of aPKC in endothelial cells. (Page 17, line 363 to page 18, 375).

2) In the abstract they refer to aPKC as an oncogene, but provide no support for this claim in the introduction, and although it is often over-expressed in tumors and mislocalized the data supporting a function as a bona fide oncogene is at best circumstantial (the one cited paper in Cancer Res claiming aPKC is an oncogene in NSCLC is just correlative).

The word oncogene was removed from the abstract.

3) Although in this study the aPKC pseudosubstrate kinase inhibitor seems to be working through aPKC, the authors should acknowledge that it is quite nonspecific. Indeed, its role in inhibiting a splice variant of aPKC (PKM) in the brain to block memory formation has been discredited.

We have included this information about the aPKC pseudosubstrate in the paper (Page 18 Line 378 to 383).

4) In Figure 2f the left and right panels do not seem to correspond – there is abundant red staining in the right but not the left panel. Perhaps the colors were switched? Also, in Figure 2e the staining is not very convincing (Right panel)

The switched labeling in Figure 2F has been corrected and the brightness of the green channel was adjusted in Figure 2e to allow for better interpretation of the merged image.

5) Figure 5 – why is the c-Myc staining not localized to the nuclei here? Is most of the staining nonspecific?

In this study we used an antibody that was raised against the complete human c-Myc antigen (Millipore 06-340) for immunostaining. It has recently been appreciated that c-Myc can be cleaved by calpain and that a cytoplasmic form of c-Myc (Myc-nick) is present in cells under different conditions (Conacci-Sorrell et al. Cell 2010; Conacci-Sorrell et al. Genes and Development 2014; Anderson et al. PNAS 2016). Myc-nick has been shown to only be detectable with specific antibodies recognizing more N-terminal regions of c-Myc and not the popular c-Myc antibody 9E10, which recognizes a c-terminal epitope and only recognizes nuclear localized c-Myc (Conacci-Sorrell et al. Genes and Development 2014). Since our antibody was also able to detect nuclear localized c-Myc under all conditions, including the images in Figure 5 (Figure 1, Supplementary Fig. 7) we believe that the non-nuclear signal is specific and may represent Myc-nick.

6) Figure 6 – what are the values of the x-axis? (months? Weeks?)

We apologize for this mistake and the labels have now been included.

7) Figure 7 b, d – the y axes are not labeled.

This has now been fixed.

8) Figure S6a – the podocalyxin looks to be cytosolic, not apical; and why is the collagen IV staining also apical, especially in the aPKC KO image?

We appreciate how carefully this reviewer examined our data. This misunderstanding was created by poor image selection and presentation of the accumulated merged image. We have now included new images that clearly display apically localized podocalyxin and basally expressed collagen IV (Supplementary Fig. 6a).

REVIEWERS' COMMENTS:

Reviewer #1 (Remarks to the Author):

In the revised manuscript authors made efforts to address most of points I raised. As a result it has improved. I have only one remaining issue in regard to the previously raised point#3. In my opinion the loss of function analysis for FOXO1 is critical.

Authors utilized a commercial product [GE Healthcare Dharmacon inc. SMART-pool ON-TARGET plus FOXO1 siRNA reagent]. This must be a proprietary product without sequence information. I suspect the effect authors observed might be due to off-target effect. It makes sense to use either lentivirus mediated FOXO1 knockdown or use a couple of distinct sequence validated siRNAs to make the point. Again, this does not stand outside the scope of the current work. The result of FOXO1 knockdown work should significantly strengthen the overall conclusion.

Reviewer #2 (Remarks to the Author):

In this revised version of the manuscript, Nakayama et al. were able to address all major issues raised by the reviewers satisfactorily. This reviewer has no further critiques.

Reviewer #3 (Remarks to the Author):

The authors have corrected the problems that were noted in the original review, particularly the labeling in Figures 2, 6, S6 and 7. They have also modified the text where appropriate. It would be helpful to show an anti-Myc blot of an angiosarcoma tissue sample, to see if the overwhelming cytosolic signal correlates with a very high abundance of Myc-nick compared to full length Myc; and show full blots of Myc in Figure S1a to determine if Myc-nick levels are altered when aPKC is silenced. Otherwise I feel that the authors have adequately addressed the issues raised in the reviews and that the manuscript is suitable for publication by Nature Communications.

We would like to thank the reviewers for their time and helpful comments. Please find below a detailed point-by-point response to all remaining questions.

Reviewer #1

Comment: In the revised manuscript authors made efforts to address most of points I raised. As a result it has improved.

Answer: We are very grateful for his/her positive assessment.

Comment: I have only one remaining issue in regard to the previously raised point#3. In my opinion the loss of function analysis for FOXO1 is critical.

Authors utilized a commercial product [GE Healthcare Dharmacon inc. SMART-pool ON-TARGET plus FOXO1 siRNA reagent]. This must be a proprietary product without sequence information. I suspect the effect authors observed might be due to off-target effect. It makes sense to use either lentivirus mediated FOXO1 knockdown or use a couple of distinct sequence validated siRNAs to make the point. Again, this does not stand outside the scope of the current work. The result of FOXO1 knockdown work should significantly strengthen the overall conclusion.

Answer: According to the editor's suggestion, we now included the sequence of the FoxO1 siRNA in Supplementary table2 which is widely used in FoxO studies (Wilhelm et al. 2016 Nature, Liu et al., Cancer Res 2008). These siRNA sequences were used to knock down FoxO1 expression over time across many different models. Therefore the effect we observed is not likely due to off-target effects.

Reviewer #2

Comment: In this revised version of the manuscript, Nakayama et al. were able to address all major issues raised by the reviewers satisfactorily. This reviewer has no further critiques.

Answer: We are very grateful for his/her positive assessment.

Reviewer #3

Comment: The authors have corrected the problems that were noted in the original review, particularly the labeling in Figures 2, 6, S6 and 7. They have also modified the text where appropriate.

Answer: We are very grateful for his/her positive assessment.

Comment: It would be helpful to show an anti-Myc blot of an angiosarcoma tissue sample, to see if the overwhelming cytosolic signal correlates with a very high abundance of Myc-nick compared to full length Myc; and show full blots of Myc in Figure S1a to determine if Myc-nick levels are altered when aPKC is silenced.

Answer: We now include full blots of Myc for Supplementary Figure 1a in Supplementary Figure 11. In this blot c-Myc was detected with the 9E10 antibody, which is known to not cross react with Myc-nick. On the other hand, c-Myc in the angiosarcoma patient detected with the c-Myc polyclonal antibody showed a double band of c-Myc corresponding to the molecular weight for full length c-Myc and Myc-nick. Importantly, this antibody was used for immunostaining of angiosarcoma patient samples, suggesting the cytosolic signal detected in the patient samples would correlate to the presence of c-Myc-nick. This information is described in page 13, line 271- page 14 line 278.